# Weighted Diversified Sampling for Efficient Data-Driven Single-Cell Gene-Gene Interaction Discovery

## Abstract

The human genome contains over 25,000 genes, yet most do not operate independently. Instead, they interact within complex networks that drive biological processes and influence intricate diseases. Understanding these gene-gene interactions is crucial but remains challenging despite advancements in experimental and computational techniques. Single-cell sequencing, which profiles millions of cells at the transcriptional level, combined with cutting-edge AI methods like Transformer-based deep neural networks, provides new opportunities to uncover subtle but critical interactions. However, Transformers' high parameter demands often hinder data efficiency, limiting their potential in large datasets. This work introduces a novel approach leveraging an advanced Transformer model to identify key gene-gene interactions. To address data efficiency challenges, we developed a weighted diversified sampling algorithm that calculates diversity scores in just two passes of the dataset. This enables efficient subset selection, allowing us to analyze only 1% of the single-cell data while maintaining performance comparable to using the full dataset. Our results highlight the power of integrating state-of-the-art AI with innovative and cost-effective sampling strategies to advance gene-gene interaction discovery, offering a scalable and efficient pathway to deeper biological insights from large-scale single-cell sequencing data.

## 1 Introduction

In the human genome, most genes function cooperatively within biological networks to execute essential processes. Within these networks, gene-gene interactions play a pivotal role in the development of complex diseases, including multiple sclerosis (Brassat et al., 2006; Motsinger et al., 2007; Slim et al., 2022), pre-eclampsia (Li et al., 2022; Diab et al., 2021; Williams & Pipkin, 2011; Oudejans & Van Dijk, 2008), and Alzheimer's Disease (Ghebranious et al., 2011; Hohman et al., 2016). Computational tools equipped with machine learning (ML) prove effective in uncovering these significant gene interactions (McKinney et al., 2006; Cui et al., 2022; Yuan & Bar-Joseph, 2021b; Wei et al., 2024; Upstill-Goddard et al., 2013). By applying ML models to large single-cell transcriptomic datasets, it is possible to uncover gene-gene interactions associated with complex, common diseases. However, many existing models depend on prior knowledge, such as transcription factors (TFs) and gene regulatory networks (Wang et al., 2019; Yuan & Bar-Joseph, 2021a; Chen et al., 2021; Shu et al., 2021) or existing gene-gene interaction (GGI) networks (Ata et al., 2020; Yuan & Bar-Joseph, 2019a), to infer new relationships. While GGI networks and TFs are invaluable for mapping biological processes, many gene-gene relationships fall outside these frameworks. Moreover, these approaches are often susceptible to high false-positive rates and biases, particularly in large-scale in vitro experiments (Mahdavi & Lin, 2007; Rasmussen & et al., 2021). To address these limitations, we propose a purely data-driven approach to uncover gene-gene interactions, eliminating reliance on prior knowledge and mitigating associated biases.

**The Rise of Transformers on Single-Cell Transcriptomic Data.** Recent advances in natural language processing, particularly the development of Transformer models (Vaswani et al., 2017), have demonstrated significant potential in biological data analysis (Hao et al., 2023; Theodoris et al., 2023; Bian et al., 2024; Cui et al., 2024). Transformer models are known for their ability to capture the dependencies between gene expressions. The information fused through the self-attention

mechanism (Vaswani et al., 2017) is particularly suited for analyzing the gene-gene relationships in single-cell transcriptomic data when modeling genes as features of a single cell. On the other hand, Transformer models also demonstrated superior performance when we scaled up their parameter size (Hao et al., 2023). This scaling capacity raises the researcher's interest in training and deploying parameter-intensive Transformer models, denoted as single-cell foundation models (Cui et al., 2024). We would like to take this advantage for better gene-gene interaction discovery by identifying feature interactions within Transformer models.

**Data-Driven Gene-Gene Interaction via Attention.** In this work, we would like to advance the gene-gene interaction discovery with the Transformer models that have demonstrated superior performance on single-cell transcriptomic data. We see the self-attention mechanism (Vaswani et al., 2017) as a pathway to facilitate the modeling of gene-gene interactions. In single-cell foundation models, the input to the model is a bag of $m$ gene expressions for a single cell. Next, in each layer and each head of the Transformer, there will be an attention map with shape $m \times m$ generated for this cell. Each entry of this attention map represents the interaction between two genes in this layer and this head. Assuming that we have a perfect Transformer that takes a cell's gene expression profile and correctly predicts if it is from a diseased tissue, we view the attention map of this cell as a strong indicator of disease-oriented gene-gene interactions.

**Efficiency Challenge in Data Ingestion.** Despite the transformative capabilities of Transformer models, one significant challenge remains: the efficient ingestion and processing of massive volumes of single-cell transcriptomic data. We are utilizing Transformer models with parameter sizes that exceed the hardware capacity, particularly that of the graphics processing unit (GPU). As a result, given a pre-trained Transformer, we have to perform batch-size computation on a massive single-cell transcriptomic dataset for computing gene-gene interactions through attention maps. This batch-size computation significantly enlarges the total execution time for scientific discovery. Moreover, the hardware in the real-world deployment environment for gene-gene interaction detection may have even more limited resources. Therefore, the current computational framework cannot support gene-gene interaction discovery on real-world single-cell transcriptomic datasets.

**Our Proposal: Two-Pass Weighted Diversified Sampling.** In this paper, we introduce a novel weighted diversified sampling algorithm. This randomized algorithm computes the diversity score of each data sample in just two passes of the dataset. The proposed algorithm is highly memory-efficient and requires constant memory that is independent of the cell dataset size. Our theoretical analysis suggests that this diversity score estimates the density of the Min-Max kernel defined on the cell-level gene expressions, which provides the foundation and justification of the proposed strategy. Through extensive experiments, we demonstrate how the proposed sampling algorithm facilitates efficient subset generation for interaction discovery. The results show that by sampling a mere 1% of the single-cell dataset, we can achieve performance comparable to that of utilizing the entire dataset.

**Our Contributions.** We summarize our contributions as fellows.

- We present a computational framework designed to advance the data-driven discovery of significant gene-gene interactions. At its core is **CelluFormer**, a Transformer-based model trained on single-cell transcriptomic data. By leveraging the Transformer's attention mechanism, CelluFormer captures complex gene-gene interactions, offering novel insights into Alzheimer's Disease.
- We pinpoint the challenge in data ingestion for the data-driven gene-gene interaction. Moreover, we argue that we should perform diversified sampling that selects a representative subset of single-cell transcriptomics data to fulfill the objective.
- We develop a diversity score for every cell in the dataset based on the Min-Max kernel density. Moreover, we perform a randomized algorithm that efficiently estimates the Min-Max kernel density for each cell. Furthermore, we use the estimated density to generate an effective subset for gene-gene interaction.

## 2 DATA-DRIVEN SINGLE-CELL GENE-GENE INTERACTION DISCOVERY

In this section, we propose a computing framework to perform gene-gene interaction discovery on single-cell transcriptomic data. We start by introducing the format of single-cell transcriptomic data. Next, we propose the formulation of our CelluFormer model tailored to single-cell data. Next, we present our multi-cell-type training to build an effective transformer model on single-cell data. Finally, given a pre-trained transformer, we showcase how to perform gene-gene interaction discovery by analyzing the attention maps.

## 2.1 Single-Cell Transcriptomic Data

Single-cell transcriptomic is a technology that profiles gene expression at the individual cell level. The profiled results, namely single-cell transcriptomic data, provide a unique landscape of gene expressions. In contrast to traditional bulk RNA-seq analysis, single-cell transcriptomic data allows for cell-level sequencing, which captures the variability between individual cells (Ata et al., 2020). Leveraging this high-resolution data allows scientists to gain insights into developmental processes, disease mechanisms, and cellular responses to environmental changes. The single-cell transcriptomic data can be formulated as a dataset with each sample as a set of gene expressions. We denote a single-cell transcriptomic dataset as $X$, where each cell $x \in X$ is a set $\{(i_1, v_1), (i_2, v_2), \cdots, (i_k, v_k)\}$. In this set, every tuple $(i, v)$ represents the expression of gene $i \in [V]$ with expression level $v \in \mathbb{R}$, where $V$ denotes the number of genes expressed at least one time in a cell $x \in X$. In this data formulation, single-cell transcriptomic data for each cell is represented as a set of gene expressions, with different cells expressing varying genes. Additionally, even when two cells express the same set of genes, their expression levels may differ. Our research objective is to identify gene-gene interactions within the vocabulary $V$ that drive complex biological processes and disease mechanisms.

## 2.2 CelluFormer: A Single-Cell Transformer

Here, we propose our Transformer architecture, CelluFormer, to learn gene-gene interactions within single-cell transcriptomic data. Based on the set formulation of single-cell transcriptomic data, we believe that the order of genes is arbitrary and biologically meaningless. Similar to scGPT (Cui et al., 2024), GeneFormer (Theodoris et al., 2023), and scFoundation (Hao et al., 2024), our method adopts a permutation-invariant design. We define our permutation-invariant condition as follows.

**Condition 2.1.** *Let $X$ denote a single-cell transcriptomic dataset. Given a single-cell data of cell $x \in X$, denoted as a set $\{(i_1, v_1), (i_2, v_2), \cdots, (i_k, v_k)\}$, a function $f : X \to \mathbb{R}$ should satisfy that, for any permutation $\pi$, $f(x) = f(\pi(x))$.*

We see Condition 2.1 as a fundamental difference between the proposed Transformer and the sequence Transformers (Vaswani et al., 2017) widely used in natural language processing. For sequence Transformers, we have to ingest sequential masks during the training to ensure that the current token does not interact with the future token. Additionally, during the inference, the sequence Transformer should perform a step-by-step generation for each token. As a result, the

Table 1: Performance comparison of models on neuronal cell dataset.

| Model | Training Dataset | F1 Score | Accuracy |
|---|---|---|---|
| MLP | Pax6 | 78.91 | 82.71 |
| | L5_ET | 62.02 | 73.31 |
| | L6_CT | 91.14 | 92.01 |
| | L6_IT_Car3 | 95.34 | 95.51 |
| | L6b | 86.01 | 88.76 |
| | Chandelier | 81.66 | 84.56 |
| | L5_6_NP | 89.33 | 90.42 |
| | All Neuronal Cell Types | 97.23 | 97.25 |
| CelluFormer | All Neuronal Cell Types | **98.12** | **98.12** |

sequence Transformer does not satisfy Condition 2.1. Moreover, the difference between CelluFormer and a vision Transformer (Dosovitskiy et al., 2020) is that the vision Transformer has a fixed sequence length for every input data sample. However, the number of genes expressed in each cell can vary a lot. The number of genes whose expression value can be detected by current single cell RNA sequencing technologies can vary from 2000 to 5000 in a cell. Thus, we utilize a padding mask for the downstream classification task. Additional details regarding the implementation of CelluFormer are provided in Appendix C.1.

## 2.3 Multi-Cell-Type Training of CelluFormer

We observe that there is a significant performance difference between Transformer models if we feed them with different styles of single-cell transcriptomic data. It is known that cells can be categorized into different types based on their gene expression profile and functionality. For instance, neuronal cells represent the cell types that fire electric signals called action potentials across a neural network (Levitan & Kaczmarek, 2015). Our study suggests that Transformers should be trained on single-cell transcriptomic data from various cell types to achieve better performance. We showcase an example in Table 1. We train a Transformer model to classify whether a cell is from an Alzheimer's Disease patient or a healthy individual. According to our study, CelluFormer proposed

in Section 2.2 trained on neuronal cells outperforms traditional multilayer perceptron (MLP) with downstream training on a single cell type. However, we do not see this gap when we perform training of CelluFormer on a single cell type. As a result, we see that the Transformers generally prefer massive exposure to the single-cell transcriptomic data.

## 2.4 Gene-Gene Interaction Discovery via Attention Maps

In this paper, we would like to accomplish the following objective.

**Objective 2.2** (Gene-gene interaction discovery). *Let $X$ denote a single-cell transcriptomic dataset. Let $\mathcal{V}$ denote the genes expressed in at least one $x \in X$. Let $f : X \to \mathbb{R}$ denote a permutation invariant (see Condition 2.1) CelluFormer. $f$ can successfully predict whether any $x \in X$ belongs to disease $D$. We would like to find a gene-gene pair $(v_1, v_2)$ that contributes the most to $f$'s performance in $X$. Here $v_1, v_2 \in \mathcal{V}$.*

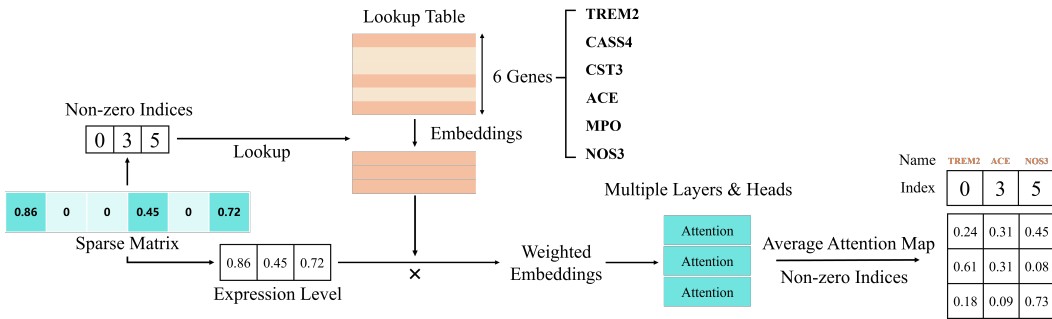

Figure 1: Gene-gene interaction modeling with attention maps.

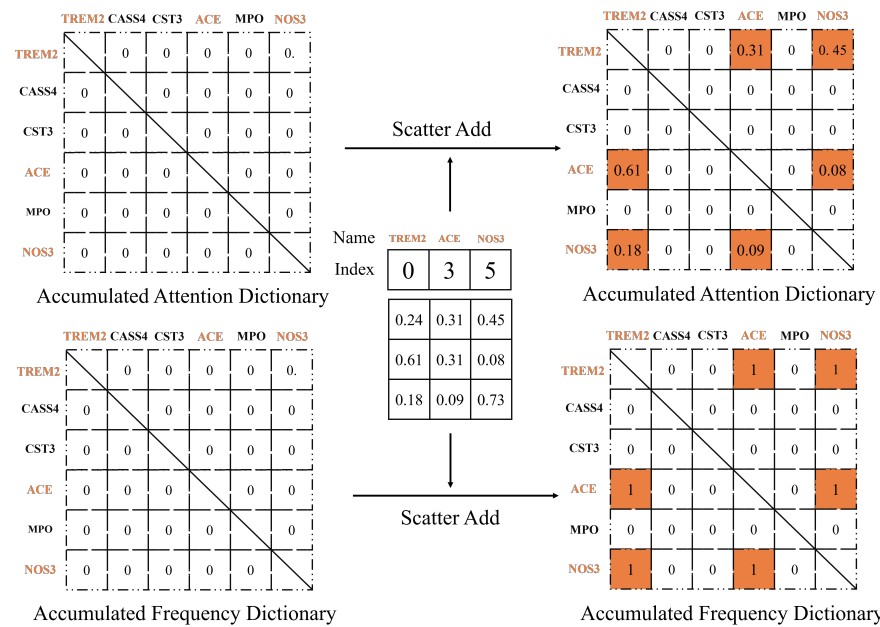

Figure 2: Accumulating multiple cells' average attention maps.

We see the self-attention mechanism of Transformers on a cell's set style gene expressions as a pathway to model gene-gene interactions. CelluFormer takes a cell $x$'s gene expressions and produces an attention map $A_{i,j} \in \mathbb{R}^{m \times m}$ at encoder block $i$ and attention head $j$. Here $m$ represents the number of genes expressed in cell $x$. Since Transformer architecture uses the Softmax function to produce $A_{i,j}$, we can view the $p$th row of $A_{i,j}$ as the interaction between gene $p$ and all other genes in

$x$. As a result, an attention map is a natural indicator of gene-gene interactions. Moreover, if we have a perfect Transformer that takes a cell $x$ gene expressions and correctly predicts if it is in a disease state, we view the attention map of this cell as an indicator of disease-oriented gene-gene interactions. Following this path, we propose a gene-gene interaction modeling approach as illustrated in Figure 1. For each cell $x$, we represent it as a set and generate a bag of embeddings from the gene embedding table. Next, we use the expression levels of each gene as a scaling factor for each gene's embedding. Next, we take the average attention maps of all layers and all heads to obtain a gene-gene interaction map in this cell.

In Objective 2.2, we would like to see not only the gene-gene interactions just for cell $x$ but also the statistical evidence of how two genes interact in the dataset $X$. As a result, we propose to accumulate multiple cells' averaged attention maps as illustrated in Figure 2. For $X$, we initialize $Z_0 \in 0^{V \times V}$ matrix as the overall attention map before aggregation and $M_0 \in 0^{V \times V}$ as the overall frequency dictionary before aggregation. Next, for each cell $x$ in the dataset, we remove its diagonal value in its averaged attention map as it represents self-interaction. Next, we perform scatter addition operations that merge $x$'s averaged attention map back to $Z_0$. We let $Z_{ij}$ add the interaction value of gene $v_i$ and $v_j$ in the average attention map of cell $x$ obtained in the Transformer model. Simultaneously, to eliminate the dataset bias of expressed genes, we count the number of appearances for each gene pair in the dataset. Once again, we perform scatter addition to record the counts back to $M_0$. This is done by updating $M_0$ through scatter addition, where $M_{ij} = M_{ij} + 1$ for every occurrence of the gene pair $(v_i, v_j)$ in the dataset. Finally, we rank the off-diagonal values in $Z$ where $Z_{ij} \leftarrow \frac{Z_{ij}}{M_{ij}}$ to retrieve the top gene-gene interaction. We note that this pipeline can be utilized with pre-trained Transformer models that have been fine-tuned on task-specific datasets, as outlined in Objective 2.2.

## 3 WEIGHTED DIVERSIFIED SAMPLING

In this section, we start by showcasing the data-efficiency problem when we use the trained Cellu-Former for gene-gene interaction discovery. Following this, we define a diversity score for each cell in the dataset and propose a two-pass randomized algorithm to efficiently compute it. Lastly, we propose a weighted diversified sampling strategy on massive single-cell data.

### 3.1 DATA-INTENSIVE COMPUTATION FOR GENE-GENE INTERACTION DISCOVERY

As illustrated in Section 2.4, once we have a pre-trained CelluFormer that can successfully predict whether a cell is in a disease state or not with its gene expressions, we can perform gene-gene interaction discovery by passing massive cells into this model and get the accumulated attention map as Figure 2. However, this process requires data-intensive computation. For every cell in the dataset, we first need to compute the average attention map as illustrated in Figure 1. Next, we perform aggregations as shown in Figure 2. It is known that CelluFormer uses plenty of trainable parameters to achieve good performance in disease state classification. As a result, the computation complexity for generating a cell's averaged attention map is expensive. Moreover, since the attention map for cell $x$ is $m \times m$, where $m$ is the number of genes expressed in $x$. Since $m$ ranges from 2000 to 5000, these giant attention maps consume the limited high bandwidth memory (HBM) in the graphics processing unit. Therefore, we have to perform batch-wise computation on a massive cell dataset for computing gene-gene interactions. Moreover, given the scale of the dataset, *any sampling algorithm with a runtime that grows exponentially with the dataset size is impractical*.

### 3.2 TWO-PASS RANDOMIZED ALGORITHM FOR COMPUTING Min-Max DENSITY

In this work, we would like to address this data-efficiency challenge by raising and asking the following research question: *Can we find a representative and small subset from the large cell dataset and still perform successful gene-gene interaction discovery?* Moreover, we would like the procedure for finding this small subset as efficient as possible.

We would like to answer this question by proposing a diversity score of a cell in the dataset. To begin with, we would like to define a kernel density on top of the Min-Max similarity between two cell's gene expressions.

---

**Algorithm 1** Two-Pass Algorithm for Estimating Min-Max Density

---

**Input:** Cell dataset $X$, 0-bit CWS function family $\mathcal{H}$ (see Definition 3.2), Hash range $B$, Rows $R$
**Output:** Min-Max density set $w$ for every $x \in X$.
**Initialize:** $A \leftarrow 0^{R \times B}$
Generated $R$ independent 0-bit CWS functions $h_1, \ldots, h_R$ from $\mathcal{H}$ with range $B$ at Random.
{We set $R = O(\log |X|)$ following the theoretical analysis of Definition 3.2}
$W \leftarrow \emptyset$
**for** $x \in X$ **do**
    **for** $r = 1 \rightarrow R$ **do**
        $A_{r,h_r(x)} + = 1$
    **end for**
**end for**
**for** $x \in X$ **do**
    **for** $r = 1 \rightarrow R$ **do**
        $w_x \leftarrow w_x + A_{r,h_r(x)}$
    **end for**
    $w_x \leftarrow w_x / R$ {$w_x$ is the estimated Min-Max density for $x$.}
    $W \leftarrow \{w_x\}$
**end for**
**return** $W$

---

**Definition 3.1** (Min-Max Density). Given a cell dataset $X \subset \mathbb{R}^V$, for every $q \in X$, we define its Min-Max density as: $\mathcal{K}(q) = \sum_{x \in X} \varphi(q, x)$, where $\varphi(q, x) : \mathbb{R} \rightarrow \mathbb{R}$ is a monotonic increasing function along with Min-Max$(q, x)$ similarity Min-Max$(q, x) = \frac{\sum_i^V \min(q_i, x_i)}{\sum_i^V \max(q_i, x_i)}$.

According to the definition, Min-Max$(x, y) \in [0, 1]$. Higher Min-Max means that two cell's gene expressions are closer to each other. Min-Max is widely viewed as a kernel (Li, 2015b; Li et al., 2021; Li & Li, 2021) in statistical machine learning. We view $\mathcal{K}(q)$ density as an indicator of how diverse $q$ is in $X$. Smaller $\mathcal{K}(q)$ means that all other $x \in X$ may be less similar to $q$, making $q$ a unique cell. On the other hand, higher $\mathcal{K}(q)$ means that $X$ has some cells that have similar gene expressions with $q$, making $q$ less unique. However, to compute $\mathcal{K}(q)$ for every $q \in X$ following Definition 3.1, we have to compute all pairwise Min-Max$(x, y)$ for any $x, y \in X$, which results in an unaffordable $O(n^2 \mathbb{NNZ}(X))$ time complexity, where $n$ is the size of $X$ and $\mathbb{NNZ}(X)$ is the maximum possible number of genes expressed in a cell $x \in X$. To reduce this $n^2$ computation, we propose a randomized algorithm that takes advantage of 0-bit consistent weighted sampling (CWS) (Li, 2015a) hash functions.

**Definition 3.2** (0-bit Consistent Weighted Sampling Hash Functions (Li, 2015a; Li et al., 2021)). Let $\mathcal{H}$ denote a randomized hash function family. If we pick a $h \in \mathcal{H}$ at random, for any two cell expressions $x, y \in \mathbb{R}^V$, we have $\Pr[h(x) = h(y)] = $ Min-Max$(x, y) + o(1)$. Here every $h \in \mathcal{H}$ is a hash function that maps any $x \in X$ to an integer in $[0, B)$. We denote $B$ as the hash range.

Here the $o(1)$ is a minor additive term with complex form. For simplicity, we refer the readers to (Li et al., 2021), Theorem 4.4 for more details.

This work presents an efficient randomized algorithm that estimates Min-Max density $\mathcal{K}(q)$ (see Definition 3.1) for every $q \in X$. As showcased in Algorithm 1, we initialize an array $A$ with all values as zeros. Next, we conduct a pass over $X$. In this pass, for every $x \in X$, we compute its hash values after $R$ independent hash functions. Next, we increment $A_{r,h_r(x)}$ with 1. After this pass, we take another pass at the dataset, for every $x \in X$, we take an average over the $A_{r,h_r(x)}$ and build a density score $w_x$. We would like to highlight that Algorithm 1 requires only two linear scans of the dataset. The time complexity for this algorithm is $O(n \mathbb{NNZ}(X))$, which is linear to the dataset. Moreover, we show that Algorithm 1 produces an estimator to Min-Max density.

**Theorem 3.3** (Min-Max Density Estimator, informal version of Theorem B.1). *Given a cell dataset $X$, for every $q \in X$, we compute $w_q$ following Algorithm 1. Next, we have $\mathbb{E}[w_q] = \sum_{x \in X}($Min-Max$(x, q) + o(1))$, where Min-Max is the Min-Max similarity defined in Definition 3.1. As a result, $w_q$ is an estimator for Min-Max density $\mathcal{K}(q)$ defined in Definition 3.1 with $\varphi(q, x) = $ Min-Max$(x, q) + o(1)$.*

We provide the proof of Theorem 3.3 in the supplementary materials.

## 3.3 Weighted Diversified Sampling with Inverse Min-Max Density

We propose to use the inverse form of Min-Max density in Definition 3.1 as a score for diversity. We define it as normalized inverse Min-Max density as below.

**Definition 3.4** (Inverse Min-Max Density (IMD)). Given a cell dataset $X$, for every $q \in X$, we define its normalized inverse Min-Max density as $\mathcal{I}(q) = \mathsf{Softmax}(1/\mathcal{K}(q))$, where $\mathcal{K}(q)$ is the Min-Max diversity for $q$ in Definition 3.1, $\mathsf{Softmax}$ is the softmax function that takes over all cells in $X$.

We view the IMD $\mathcal{I}(q) \in [0, 1]$ as a monotonic increasing function for the diversity of $q$. Higher $\mathcal{I}(q)$ means that all other $x \in X$ may be less similar to $q$, making $q$ a unique cell. Moreover, IMD can be directly used as a sample probability to generate a representative subset of $X$ for Objective 2.2. Given $X$, we perform sampling without replacement to generate a subset $X_{\mathsf{sub}} \subset X$, where $x \in X$ has the sampling probability $\mathcal{I}(x)$. The advantages of sampling with IMD (see Definition 3.4) can be summarized as follows.

- The IMD $\mathcal{I}(q)$ can be an effective indicator for how diverse $q$ is in dataset $X$.
- Computing IMD is an efficient one-shot preprocessing process with just two linear scans of $X$ with time complexity $O(n\mathbb{NNZ}(X))$, where $n$ and $\mathbb{NNZ}(X)$ is defined in Section 3.2.
- The memory complexity of computing IMD is $O(RB)$, which can be viewed as constant since it is independent of $n$ and $\mathbb{NNZ}(X)$.

In the following definition, we would like to estimate the interaction score with WDS. Moreover, we show that WDS serves an unbiased estimator of the interaction score obtained from the whole dataset. This unbiased estimator builds on the theoretical analysis of local density estimation Wu et al. (2018). We suggest the ideal sample size to estimate the target interaction score with a multiplicative error of $\varepsilon$ and failure probability $\delta$ bounded by $O(\log^2(n) \cdot \log(1/\delta)/\varepsilon^2)$, where $n$ represents the total number of elements in the dataset.

**Definition 3.5** (Estimated Interaction Score with WDS). Let $Z_x(v_i, v_j)$ denote the interaction value of gene $v_i$ and $v_j$ in the average attention map of cell $x$ obtained in the CelluFormer. For dataset $X$, we perform a sampling where each cell $x \in X$ is sampled with probability $\mathcal{I}(x)$ (see Definition 3.4) and get a subset $X_s$. Next, we define the estimated interaction score between gene $v_i$ and $v_j$ learned from $X$ as:

$$\widetilde{Z}(v_i, v_j) = \frac{\sum_{x \in X_s} Z_x(v_i, v_j) \cdot \mathcal{I}(x)}{\sum_{x \in X_s} \mathcal{I}(x)},$$

where $\widetilde{Z}(v_i, v_j)$ is an unbiased estimator for the expectation of $Z(v_i, v_j)$ in distribution with density $\mathcal{I}(x)$. Formally,

$$\mathbb{E}[\widetilde{Z}(v_i, v_j)] = \mathbb{E}_{x \sim \mathcal{I}(x)}[Z_x(v_i, v_j)],$$

$$\mathbf{Var}[\widetilde{Z}(v_i, v_j)] = \frac{\sum_{x \in X_s} \mathcal{I}(x)^2}{(\sum_{x \in X_s} \mathcal{I}(x))^2} \mathbf{Var}_{x \sim \mathcal{I}(x)}[Z_x(v_i, v_j)].$$

## 4 Experiment

In this section, we want to validate the effectiveness of our gene-gene interaction pipeline as well as the two-pass diversified sampling algorithm 1. There are a few research questions we want to answer:

- **RQ1:** How does the proposed Transformer-based computing framework introduced in Section 2 perform in gene-gene interaction discovery?
- **RQ2:** How does the Min-Max density estimated by two-pass diversified sampling Algorithm 1 characterize the diversity of a cell in the whole dataset? Is this estimated Min-Max density useful?
- **RQ3:** How does the estimated Min-Max density perform in improving data-efficiency of gene-gene interaction discovery? How is the quality of the subset sampled according to the estimated Min-Max density?

## 4.1 SETTINGS

**Dataset:** For the training dataset, we employ the Seattle Alzheimer's Disease Brain Cell Atlas (SEA-AD) (Gabitto et al., 2023), which includes single nucleus RNA sequencing data of 36,601 genes (as 36,601 features) from 84 senior brain donors exhibiting varying degrees of Alzheimer's Disease (AD) neuropathological changes as well as healthy control. By providing extensive cellular and genetic data, SEA-AD enables in-depth exploration of the cellular heterogeneity and gene expression profiles associated with AD. To facilitate a comparative analysis between AD-affected and non-AD brains, we select cells from 42 donors classified within the high-AD category and 9 donors from the non-AD category, based on their neuropathological profiles. This selection criterion ensures a robust comparison, aiding in the identification of gene-gene interactions linked to AD progression (Gabitto et al., 2023). The dataset is comprehensively annotated, covering 1,240,908 cells across 24 distinct cell types. The labels of the cell types are provided by the data generator. Our analysis focused on several types of neuron cells as they are most relevant to AD – a neural degenerative disease. We selected 18 neuronal cell types as our final training dataset since we believe neuronal cells are more likely to reveal explainable gene-gene interactions that are related to Alzheimer's Disease compared to non-neuronal cells. To better detect expression relationships among genes, we apply the Seurat Transformation Function (Stuart et al., 2019) to eliminate the problem of sequence depth difference.

**Model:** For the SEA-AD dataset, we designed a CelluFormer model as explained in 2.2 to predict labels indicative of AD conditions. Further details can be found in the Appendix C.1.

**Baselines:** Our proposed algorithm leverages the attention maps of the Transformer models. As a result, we can apply this algorithm to existing pre-trained single-cell Transformers, e.g. scGPT Cui et al. (2024) and scFoundation Hao et al. (2024) to perform gene-gene interaction. Additionally, we compare our method with three statistical methods, Pearson Correlation, CS-CORE, and Spearman's Correlation (Freedman et al., 2007; Su et al., 2023; De Smet & Marchal, 2010). While these methods are widely adopted by biologists for gene co-expression analysis, gene co-expression values alone do not provide information about the relationship between gene pairs and Alzheimer's Disease. To identify gene-gene interactions relevant to Alzheimer's Disease, we apply these methods to subsets containing disease and non-disease cells respectively, and calculate their gene co-expression values. The difference in co-expression values between disease and non-disease cells is then used as a final score to rank the gene pairs. We also present more experiments in Appendix D.1 that demonstrate how Transformers aggregate data with varying labels.

Our baseline includes NID (Tsang et al., 2017), a traditional feature interpretation technique that extracts learned interactions from trained MLPs. NID identifies interacting features by detecting strongly weighted connections to a standard hidden unit in MLPs after training. We evaluated our CelluFormer model against the MLP model, with performance results presented in Table 1.

Additionally, to comprehensively evaluate RQ1, we utilized two existing single-cell large foundation models to assess our algorithm. Specifically, we fine-tuned two foundation models, scFoundation (Hao et al., 2024) and scGPT (Cui et al., 2023), to classify whether a cell is AD or non-AD (performance results are provided in Table 4). We then applied our gene-gene interaction discovery pipeline using the attention maps of these foundation models. In the sampling experiments, we compare WDS with uniform sampling since none of them requires preprocessing time exponential to the dataset size.

**Evaluation Metric:** For a comprehensive evaluation encompassing the entire ranked list of gene-gene interactions, we utilized the Kolmogorov-Smirnov test, which was facilitated by the `GSEApy` package (Fang et al., 2023) in Python. We select normalized enrichment score (NES) (Subramanian et al., 2005) as our evaluation metric. The ground truth dataset is sourced from *BioGRID* and *DisGenet* (Oughtred et al., 2019; Piñero et al., 2016). For our experiments, we extract a subset of DisGenet that includes genes associated with Alzheimer's Disease. We then filter out genes in BioGRID that are not present in this DisGenet subset. Finally, we obtain a filtered BioGRID dataset containing only genes relevant to Alzheimer's Disease. We provide more explanations about our evaluation metrics in Appendix C.2.

## 4.2 THE EFFECTIVENESS OF TRANSFORMERS IN GENE-GENE INTERACTION DISCOVERY (RQ1)

We evaluate our gene-gene interaction discovery framework across seven distinct cell types to assess its performance comprehensively. The results, summarized in Table 2, compare the proposed framework applied to Transformer-based models, including Celluformer, scGPT, and scFoundation.

For comparison, Table 2 also presents results from non-Transformer deep neural network baselines, such as NID, and traditional non-deep learning methods, including Pearson, CS-CORE, and Spearman. The findings demonstrate that the proposed Transformer-based framework significantly improves the effectiveness and stability of gene-gene interaction extraction. Moreover, among the Transformer-based models, Celluformer consistently achieves superior performance compared to scGPT and scFoundation. The performance of foundation models like scGPT and scFoundation may stem from various factors. For instance, the data handling approaches of foundation models, such as using rank instead of absolute expression values in scGPT, combined with the vast datasets used for training, make it challenging to isolate all factors contributing to the observed lower performance. We hypothesize that the potential influences may include differences in gene vocabulary and model training dynamics. To better understand these factors and their influence on model performance, particularly in identifying gene-gene interactions, future research should include a thorough evaluation.

Table 2: Performance comparison of models on neuronal cell data. To evaluate different models on datasets with varying sizes, we further select 7 neuronal cell types from all neuronal cell types. CelluFormer, scGPT, scFoundation, MLP, Pearson Correlation, Spearman's Correlation, and CS-CORE were tested on 8 different datasets to obtain their gene pair rankings.

| Dataset | CelluFormer | scFoundation | scGPT | NID | Pearson | CS-CORE | Spearman |
|---|---|---|---|---|---|---|---|
| L5_ET | 1.15 | 1.04 | **1.23** | 0.90 | 0.50 | 1.11 | 0.91 |
| L6_CT | 1.18 | 1.03 | 1.17 | **1.54** | -0.21 | 1.06 | 0.72 |
| Pax6 | **1.25** | 0.82 | 1.01 | 1.04 | 0.93 | 0.95 | 1.15 |
| L5_6_NP | 1.21 | 1.06 | **1.50** | 1.49 | 0.87 | 0.92 | 0.95 |
| L6b | 1.13 | 0.99 | **1.23** | 0.62 | 0.75 | 0.62 | 1.08 |
| Chandelier | **1.17** | 1.16 | 1.09 | 1.07 | 0.94 | 1.06 | 0.96 |
| L6_IT_Car3 | **1.22** | 0.90 | 0.66 | 1.19 | 0.59 | 1.08 | 0.86 |
| All neuron data | **1.17** | 1.02 | 0.99 | 0.86 | 1.01 | 1.06 | 1.04 |

## 4.3 Ablation Studies (RQ2 & RQ3)

We addressed these questions by comparing our weighted diversified sampling (WDS) method with uniform sampling across various sample sizes, ranging from 1% to 10% of the original dataset. We generated data subsets for each cell type using WDS and uniform sampling. We then applied our Transformer-based framework for feature selection at each sample size. Since CelluFormer consistently outperformed other baselines, we selected it as our base model. We repeated each experiment five times and recorded the NES scores as the results. To evaluate the sampling methods, we calculated the average NES score across the five experiments. We also computed the Mean Square Error (MSE) between the NES scores from the sampling experiments and the ground truth derived from the entire dataset, as shown in Table 2. The evaluation results are presented in Table 3. We note that WDS consistently produced higher NES scores compared to uniform sampling. As the sample size increased, the NES scores from uniform sampling began to converge with the ground truth. In contrast, the NES scores from WDS consistently remained close to the ground truth, even at smaller sample sizes. The result indicates that while WDS offers a significant advantage in small samples by enabling the Transformer to capture a broader range of genetic interactions, its benefits diminish as more data becomes available. We also provide a detailed study on the choice of parameter $R$ in Algorithm 1 in Appendix D.2.

## 5 Related Work

**Single-Cell Transformer Models.** Single-cell RNA sequencing (scRNA-seq), or single-cell transcriptomics, enables high-throughput insights into cellular systems, amassing extensive databases of transcriptional profiles across various cell types for the construction of foundational cellular models (Hao et al., 2023). Recently, there has emerged a large number of transformer models pre-trained for single-cell RNA sequencing tasks, including scFoundation (Hao et al., 2023), Geneformer (Theodoris et al., 2023), scMulan (Bian et al., 2024), scGPT (Cui et al., 2024). These foundation models have gained a progressive understanding of gene expressions and can build meaningful gene encodings over limited transcriptomic data. Yet, the previous work did not pay attention to pairwise gene-gene interactions. In our work, we would like to highlight a fundamental functionality of single-cell foundation models: we must use these models to perform data-driven scientific discovery.

Table 3: Evaluation Results for the transformer over sample data. For each cell type, we performed 8 groups of down-sampling regarding 4 different sample sizes and 2 sampling methods. We let the transformer conduct inferences over the sample data and generate results.

| Dataset | Sample Size | Mean of NES | | MSE of NES | |
|---|---|---|---|---|---|
| | | Uniform | WDS | Uniform | WDS |
| L5_ET | 1% | 0.90 | **0.95** | 0.0127 | **0.0082** |
| | 2% | 0.89 | **1.17** | 0.0131 | **0.0001** |
| | 5% | 1.02 | **1.19** | 0.0036 | **0.0003** |
| | 10% | 0.87 | **1.07** | 0.0158 | **0.0012** |
| L6_CT | 1% | 0.85 | **1.19** | 0.0207 | **0.0000** |
| | 2% | 1.05 | **1.18** | 0.0030 | **4.30e-05** |
| | 5% | 0.93 | **1.23** | 0.0122 | **0.0006** |
| | 10% | 0.91 | **1.21** | 0.0136 | **0.0002** |
| Pax6 | 1% | 0.94 | **1.08** | 0.0184 | **0.0053** |
| | 2% | 1.03 | **1.18** | 0.0098 | **0.0009** |
| | 5% | 0.98 | **1.20** | 0.0139 | **0.0004** |
| | 10% | 1.06 | **1.17** | 0.0072 | **0.0012** |
| L5_6_NP | 1% | 0.90 | **1.13** | 0.0192 | **0.0016** |
| | 2% | **1.15** | 1.11 | **0.0009** | 0.0021 |
| | 5% | 1.02 | **1.20** | 0.0076 | **4.54e-06** |
| | 10% | 1.01 | **1.17** | 0.0080 | **0.0004** |
| L6b | 1% | 0.79 | **1.17** | 0.0226 | **0.0004** |
| | 2% | 0.76 | **1.14** | 0.0266 | **0.0000** |
| | 5% | 0.88 | **1.20** | 0.0121 | **0.0009** |
| | 10% | 1.20 | **1.21** | **0.0010** | 0.0014 |
| L6_IT_Car3 | 1% | 0.78 | **1.20** | 0.0384 | **0.0001** |
| | 2% | 0.87 | **1.15** | 0.0242 | **0.0011** |
| | 5% | 0.97 | **1.17** | 0.0123 | **0.0006** |
| | 10% | 0.97 | **1.18** | 0.0123 | **0.0003** |

**Randomized Algorithms for Efficient Kernel Density Estimation.** Kernel density estimation (KDE) is a fundamental task in both machine learning and statistics. It finds extensive use in real-world applications such as outlier detection (Luo & Shrivastava, 2018; Coleman et al., 2020) and genetic abundance analysis (Coleman et al., 2022). Recently, there has been a growing interest in applying hash-based estimators (HBE)(Charikar & Siminelakis, 2017; Backurs et al., 2019; Siminelakis et al., 2019; Coleman et al., 2020; Spring & Shrivastava, 2021) for KDE. HBEs leverage Locality Sensitive Hashing (LSH)(Indyk & Motwani, 1998; Datar et al., 2004; Li et al., 2019) functions, where the collision probability of two vectors under an LSH function is monotonic relative to their distance measure. This property allows HBE to perform efficient importance sampling using LSH functions and hash table-type data structures. Furthermore, (Liu et al., 2024) extend KDE algorithms as a sketch for the distribution. However, previous works have not considered LSH for weighted similarity. In this work, we focus on designing a new HBE that incorporates the Min-Max similarity (Li, 2015b), a weighted similarity measure.

## 6 CONCLUSION

Gene-gene interactions are pivotal in the development of complex human diseases, yet identifying these interactions remains a formidable challenge. In response, we have developed a pioneering approach that utilizes an advanced Transformer model to effectively reveal significant gene-gene interactions. Although the Transformer models are highly effective, their extensive parameter requirements often impede efficient data processing. To overcome this limitation, we have introduced a weighted diversified sampling algorithm. This innovative algorithm efficiently calculates the diversity score of each data sample across just two passes of the dataset. With this method, we enable the rapid generation of optimized data subsets for interaction analysis. Our comprehensive experiments illustrate that by leveraging this method to sample a mere 1% of the single-cell dataset, we can achieve results that rival those obtained using the full dataset, significantly enhancing both efficiency and scalability.

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

APPENDIX

## A  MORE RELATED WORK ON GENE-GENE INTERACTION DISCOVERY

In this section, we provide a more detailed review of the existing work on gene-gene interaction discovery. There exists a series of machine learning approaches for gene-gene interaction. However, we argue that these existing works do not directly identify gene-gene relationships from the single-cell RNA data. Instead, they frame the gene-gene relationships into prior biological concepts. For instance, the goal of SCENIC (Aibar et al., 2017), GRNBoost2 (Moerman et al., 2018), SCODE(Matsumoto et al., 2017), and SCRIBE(Qiu et al., 2020) is to discover gene regulatory network (GGI) (Vanunu et al., 2010; Erten et al., 2011; Chen et al., 2021b; Yuan & Bar-Joseph, 2019b), with the explicit intension of using transcription factor-target gene concept framework to model the data for gene regulation discovery. VGAE (Singh & Lio', 2019) and GCAS (Rao et al., 2018) explore the potential to incorporate GNN and auto-encoder structure in the GGI network. Additionally, multiple existing works utilize machine learning models such as SVMs for gene-gene interaction discovery Shen et al. (2010); Matchenko-Shimko & Dube (2007); Chen et al. (2008). However, these methods are studying single nucleotide polymorphisms (SNPs) data instead of targeting single-cell RNA data (Uffelmann et al., 2021) In contrast, our method takes a data-driven approach to identify gene-gene relationships without framing such relationships into any biological concepts like GGIs. Even though gene regulation is an important gene-gene relationship from the transcription profile, there could be other subtle signals of gene-gene interaction beyond gene regulation. Therefore, the scope and conceptual framework of our work are different from those works.

## B  PROOFS OF THEOREM 3.3

**Theorem B.1** (Min-Max Density Estimator, formal version of Theorem 3.3). *Given a cell dataset $X$, for every $q \in X$, we compute $w_q$ following Algorithm 1. Next, we have*

$$\mathbb{E}[w_q] = \sum_{x \in X} (\mathsf{Min\text{-}Max}(x, q) + o(1)),$$

*where* Min-Max *is the* Min-Max *similarity defined in Definition 3.1. As a result, $w_q$ is an estimator for* Min-Max *density $\mathcal{K}(q)$ defined in Definition 3.1 with $\varphi(q, x) = \mathsf{Min\text{-}Max}(x, q) + o(1)$.*

*Proof.* According to Theorem 2 in (Coleman et al., 2019), the expectation of $w_q$ should be:

$$\mathbb{E}[w_q] = \sum_{x \in X} \Pr_{h \sim \mathcal{H}}[h(q) = h(x)]$$

According to Definition 3.2, we have

$$\Pr_{h \sim \mathcal{H}}[h(q) = h(x)] = \mathsf{Min\text{-}Max}(x, q) + o(1)$$

.

As a result,

$$\mathbb{E}[w_q] = \sum_{x \in X} (\mathsf{Min\text{-}Max}(x, q) + o(1))$$

Moreover, since $\mathsf{Min\text{-}Max}(x, q) + o(1)$ is a monotonic increasing function of $\mathsf{Min\text{-}Max}(x, q)$. We say that $w_q$ is an estimator for Min-Max density $\mathcal{K}(q)$ defined in Definition 3.1 with $\varphi(q, x) = \mathsf{Min\text{-}Max}(x, q) + o(1)$. □

## C  EXPERIMENT DETAILS

### C.1  MODEL IMPLEMENTATIONS

**Transformer Configurations:** In this work, we used the standard multi-head self-attention introduced in (Vaswani et al., 2017). We do not see the potential of the proposed blocks in (Lee et al., 2019) in

our setting. Moreover, we perform padding on each batch of training and inference of single-cell data. Accordingly, we introduce a padding mask in the attention mechanism to avoid computation on the padded position. For each input sequence, we represent them as embedding by a lookup table that maps a vocabulary of 36,601 genes to 128-dimensional vectors. Subsequently, the embedded data passes through 4 transformer encoder blocks. Each encoder block features 8 attention heads, to capture complex, non-linear relationships within the data. Finally, the output is fed into a linear layer that classifies the data labels. Here the label for the cell can be disease-oriented, such as whether this cell is from an Alzheimer's disease patient. We represent each input sequence by employing a lookup table that transforms a comprehensive vocabulary of 36,601 genes into 128-dimensional embedding vectors. These vectors are subsequently processed through a series of 4 Transformer encoder blocks. Each encoder block is equipped with 8 attention heads, a 512-dimensional feedforward layer, and a dropout layer in a ratio of 0.1. The processed outputs are then directed to a linear classification layer, which is tasked with predicting labels indicative of Alzheimer's Disease conditions. We adopted the Adam Optimization Algorithm to minimize the loss function Kingma & Ba (2017). The model is trained under a learning rate of 1e-5 and the batch size of our data-loader is set as 128. The testing results for the transformer after 3 epochs of training are given in Table 1.

**MLP Configurations:** The MLP consists of 2 hidden layers, with 128 and 256 hidden units respectively. Each hidden layer is followed by a dropout and a Softplus module. The MLP is trained under a learning rate of 1e-4 and the batch size of our data-loader is set as 128. We adopted the Adam Optimization Algorithm to minimize the loss function Kingma & Ba (2017). The testing results for the MLP after 80 epochs of training are given in Table 1.

Table 4: Complete Performance comparison of models on neuronal cell data.

| Model | Training Dataset | F1 Score | Accuracy |
|---|---|---|---|
| MLP | Pax6 | 78.91 | 82.71 |
| | L5_ET | 62.02 | 73.31 |
| | L6_CT | 91.14 | 92.01 |
| | L6_IT_Car3 | 95.34 | 95.51 |
| | L6b | 86.01 | 88.76 |
| | Chandelier | 81.66 | 84.56 |
| | L5_6_NP | 89.33 | 90.42 |
| | All Neuronal Cell Types | 97.23 | 97.25 |
| CelluFormer | All Neuronal Cell Types | **98.12** | **98.12** |
| scGPT | All Neuronal Cell Types | 93.85 | 94.32 |
| scFoundation | All Neuronal Cell Types | 97.38 | 97.39 |

**Fine-tuning configurations for scFoundatoin and scGPT:** For fine-tuning scGPT, we use an LR of 1e-4 and a batch size of 64. We utilize a step scheduler down to 90% of the original learning rate every 10 steps. The training process converges after 6 epochs. For scFoundation, we use an LR of 1e-4 and a batch size of 32. We fine-tune scFoundation for 10 epochs. The performances of scFoundation and scGPT on classifying disease cells are shown in Table 4.

**Implementation and Computation Resources:** Our codebase and workflow are implemented in PyTorch Paszke et al. (2017). We trained and tested our workflow on a server with 8 Nvidia Tesla V100 GPU and a 44-core/88-thread processor (Intel(R) Xeon(R) CPU E5-2699A v4 @ 2.40GHz).

## C.2 EVALUATION METRICS

The normalized enrichment score (NES) is the main metric used to analyze gene set enrichment outcomes Subramanian et al. (2005). This score quantifies the extent of over-representation of a ground truth dataset at the top of the ranked list of gene-gene interactions. That is, the higher the better. We can calculate NES by starting at the top of the ranked list and moving through it, adjusting a running tally by increasing the score for each gene-gene interaction in the ground truth dataset and decreasing it for others based on each gene-gene interaction's rank. This process continues until we evaluate the entire ranked list to identify the peak score, which is the enrichment score. The BioGRID Dataset provides human protein/genetic interactions. Specifically, *BioGRID* contributes $204,831$

protein/genetic interactions that help verify the enrichment of genuine biological interactions in a ranked list of gene-gene interactions. DisGenet contains 429,036 gene-disease associations (GDAs), connecting 17,381 genes to 15,093 diseases, disorders, and abnormal human phenotypes Oughtred et al. (2019); Piñero et al. (2016).

# D MORE EXPERIMENTS

## D.1 CONTRASTIVE RANKING

Here, we also explore alternative strategies for aggregating attention maps. While Pearson Correlation, Spearman's Correlation, and CS-CORE themselves cannot capture the information between gene pairs the the target disease, we believe Transformers learn the difference among data with varying labels. Hence, we do not need to calculate the difference between attention maps aggregated on data with varying labels. However, given that the Transformer is trained to classify disease cells, we hypothesize that it likely assigns significant attention to specific gene pairs within disease cells. To evaluate this, we applied our pipeline to three distinct datasets. The experimental results summarized in Table 5 show that our pipeline achieves improved NES when both disease and non-disease cells are used as inputs. These findings suggest that the Transformer benefits from data both positive and negative labels to provide a more comprehensive understanding of features.

Table 5: This experiment involves three groups. In the first group, the Transformer only takes the disease cells for inference. We directly evaluate the ranked list given by aggregated attention map across disease cells. In the second group, we calculate the aggregated attention maps on the disease cells and the non-disease cells respectively. The final attention map is obtained by subtracting these two attention maps. The third group is to aggregate attention maps across the whole dataset.

| Strategy | L5_ET | L6_CT | Pax6 | L5_6_NP | L6b | Chandelier | L6_IT_Car3 |
|---|---|---|---|---|---|---|---|
| AD cells | 1.09 | 1.09 | 0.98 | 0.78 | 1.13 | 0.90 | 0.89 |
| AD cells - Non-AD cells | 1.08 | 0.89 | 1.05 | 0.76 | 0.82 | 0.65 | **1.39** |
| All cells | **1.15** | **1.18** | **1.25** | **1.21** | **1.13** | **1.17** | 1.22 |

## D.2 EMPIRICAL STUDY ON PARAMETER $R$ IN ALGORITHM 1

Table 6: The Mean value of NES results across 5 experiments on L5_ET, L6_CT, and Pax6 cell type datasets.

| Dataset | Sample Size | Mean of NES | | | |
|---|---|---|---|---|---|
| | | Uniform | WDS with R=100 | WDS with R=200 | WDS with R=500 |
| L5_ET | 1% | 0.90 | **1.02** | 0.95 | 0.93 |
| | 2% | 0.89 | **1.17** | **1.17** | 0.97 |
| | 5% | 1.02 | 0.97 | **1.19** | 1.11 |
| | 10% | 0.87 | 1.01 | **1.07** | **1.07** |
| L6_CT | 1% | 0.85 | **1.19** | **1.19** | 1.11 |
| | 2% | 1.05 | **1.21** | 1.18 | 1.09 |
| | 5% | 0.93 | 1.13 | **1.23** | 1.21 |
| | 10% | 0.91 | **1.23** | 1.21 | 1.20 |
| Pax6 | 1% | 0.94 | **1.13** | 1.08 | 1.17 |
| | 2% | 1.03 | **1.22** | 1.18 | 1.19 |
| | 5% | 0.98 | **1.21** | 1.20 | 1.19 |
| | 10% | 1.06 | 1.19 | 1.17 | **1.22** |

During our experiments on WDS, we observed that the value of $R$ (see Algorithm 1) has a noticeable impact on NES performance. In Table 6 and Table 7, we evaluate three different $R$ values ranging from 100 to 500. The results demonstrate that increasing $R$ leads to a significant decline in NES. Although WDS with smaller $R$ values yields relatively higher NES, it tends to diverge from the NES calculated on the entire dataset.

Table 7: The MSE of NES results across 5 experiments on L5_ET, L6_CT, and Pax6 cell type datasets. The MSE values are calculated according to the results in Table 2.

| Dataset | Sample Size | MSE of NES | | | |
| --- | --- | --- | --- | --- | --- |
| | | Uniform | WDS with R=100 | WDS with R=200 | WDS with R=500 |
| L5_ET | 1% | 0.0636 | 0.0408 | **0.0178** | 0.0477 |
| | 2% | 0.0653 | 0.0005 | **0.0004** | 0.0339 |
| | 5% | 0.0181 | **0.0014** | 0.0310 | 0.0018 |
| | 10% | 0.0790 | **0.0062** | 0.0192 | 0.0064 |
| L6_CT | 1% | 0.1033 | 0.0002 | 0.0002 | 0.0046 |
| | 2% | 0.0151 | **0.0001** | 0.0014 | 0.0070 |
| | 5% | 0.0610 | 0.0028 | **0.0025** | 0.0013 |
| | 10% | 0.0681 | 0.0011 | 0.0031 | **0.0007** |
| Pax6 | 1% | 0.0920 | 0.0264 | 0.0135 | **0.0057** |
| | 2% | 0.0488 | 0.0047 | **0.0006** | 0.0027 |
| | 5% | 0.0695 | 0.0022 | **0.0015** | 0.0027 |
| | 10% | 0.0362 | 0.0058 | 0.0028 | **0.0008** |

