# OpenReview forum: "Weighted Diversified Sampling for Efficient Data-Driven Single-Cell Gene-Gene Interaction Discovery"
_ICLR.cc/2025/Conference — Submitted to ICLR 2025_

### Official Review · Reviewer_8Hw9 · 2024-10-31

**Soundness:** 2
**Presentation:** 2
**Contribution:** 1
**Rating:** 1
**Confidence:** 4

**Summary:**

This paper aims to use transformer model trained from scRNA-seq data to identify gene-gene interactions.  The approach involves combining the attention matrices over all the layers of the transformer and then evaluating whether the resulting aggregated attention values give higher weights to known pairs of interacting genes.  To make the approach computationally feasible, the authors also develop a sketching procedure to select a representation subset of cells.

**Strengths:**

The problem of detecting gene-gene interactions directly from scRNA-seq data is important.

**Weaknesses:**

The main idea here -- that you can infer gene-gene interactions by looking at the attention map in the transformer -- is pretty obvious.

The proposed sketching procedure is not compared to any existing methods.

The proposed transformer model is insufficiently described, and the paper doesn't say how it differs from existing models.

One of the major contributions here is a method for finding representative subsets of cells from scRNA-seq data (Section 3).  Unfortunately, this problem is already fairly well studied, and the paper fails to cite any existing methods that tackle this problem (e.g., Hie et al., Cell Systems 2019; Yang et al., ACM-BCB 2020; Yi & Stanley, bioRxiv, 2023; Hao et al. Nat Biotech 2024).  These methods should be employed and compared against.

More generally, there are many existing methods for detecting gene-gene interactions from scRNA-seq data.  Prominent examples include SCENIC (Aibar Nature Methods 2017), GRNBoost2 (Moerman Bioinformatics 2019), PIDC (Chan Cell Systems 2017), SCODE (Matsumoto Bioinformatics 2017), SCRIBE (Moerman Nature Communications 2019).  This large literature is not cited, and none of the methods therein is compared against.

In line 161, you say that you train a Transformer model to classify whether a cell is an "Alzheimer's disease-infected cell or not."  First, Alzheimer's is not an infection.  But more importantly, there is no way to label individual cells as being affected by Alzheimer's or not.  I am guessing that this sentence should say that you are labeling cells based on whether they come from an individual with Alzheimer's disease.  How exactly this is done should be clarified.

The evaluation of the gene-gene interactions is problematic.  The approach amounts to filtering a large set of known gene-gene interactions to include only those interactions that are implicated in Alzheimer's disease.  This ignores the fact that many genes not involved in Alzheimer's disease continue to function and interaction with one another in the cell.  It's not actually clear to me whether the evaluation considers pairs of genes not involved in Alzheimer's.  It seems, from the description, like these pairs are treated as non-interacting.

**Questions:**

Why develop a new transformer model if the goal is just to identify gene-gene interactions? It seems like it would be easier to work with some existing model.

How does the proposed Transformer model differ from scGPT, GeneFormer, and scFoundation?  On line 124 we are told that they are similar to CelluFormer, but I don't see any indication of how they differ.

How do the "scatter addition operations" described in line 231 work?  This entire paragraph would benefit from a more formal treatment, since I found the textual description very hard to follow.

On the face of it, the fact that you can achieve the same performance from 1% of the data as from 100% can have two interpretations: you made fanastic use of the 1% to achieve performance comparable to 100%, or you made terrible use of the 100% and only achieved performance as good as using 1%.  How do we know which is the case here?

How are the cell type labels in Table 1 derived? More detail needs to be given to make this experiment reproducible.

---

> ### Author Response · Authors · 2024-11-25
>
> We thank the reviewer for the comments. Please see the following clarification.
>
> # [W1 - Evaluation of sampling algorithms - Our proposed sampling algorithm is a diversified sampling approach with the same sampling complexity as uniformly random sampling.]
> >The proposed sketching procedure is not compared to any existing methods.
>
> We demonstrate in Algorithm 1 that our diversified sampling approach requires only a linear pass over the dataset, significantly reducing preprocessing time compared to other algorithms with exponential preprocessing complexity in terms of the dataset size $n$. Furthermore, once preprocessing is complete, our sampling procedure operates in $O(1)$ time, matching the efficiency of uniform sampling.
>
>
> As a result, in this paper, we focuses on how to achieve diversified sampling under the constaint that the sampling procedures operates in $O(1)$ time and the preprocessing procedures operates in $O(nd)$. We do not see other sampling algorithms based on kernel density follow this constaint. So we showcase a comparison with uniform sampling.
>
>
> # [W2 - Utilization of Transformer models - We highlight the usage of Transformer models for interaction modelling on set of tokens.]
> >The proposed transformer model is insufficiently described, and the paper doesn't say how it differs from existing models.
>
> We present two key features of Celluformer. First, the model operates on a set of gene expressions without assuming a sequential order. Consequently, Celluformer avoids introducing positional biases, such as positional encodings or causal masking, in its architecture. Second, our objective is not to use the Transformer model for performance prediction. Instead, we leverage its attention maps to infer novel gene-gene interactions, utilizing the attention mechanisms as a tool for discovery. This approach represents an empirical advancement, applying state-of-the-art Transformer architectures for scientific exploration.  This is described on **line 224-237** and **Figure 1**  in the manuscript.
>
>
> # [W3 - Study on the mentioned literature - We highlight the foundamental difference in the settings with these approaches with a study.]
> >One of the major contributions here is a method for finding representative subsets of cells from scRNA-seq data (Section 3). Unfortunately, this problem is already fairly well studied ...
>
>
> We would like to clarify our relationship with the mentioned literature.
>
> First, we would like to clarify Section 3. Section 3 is about developing a novel sampling method for data efficiency when performing gene-gene interaction discovery. To our best knowledge, this sampling approach is novel and has not been proposed or published by others.
>
> Secondly, we appreciate the reviewer’s recommendation of some existing methods in gene-gene interactions. However, there are some fundamental differences between these methods and our method:
> 1. The goal of SCENIC, GRNBoost2, SCODE and SCRIBE is to discover gene regulatory network, with explicit intension of using transcription factor-target gene concept framework to model the data for gene regulation discovery. On the other hand, our method takes a data driven approach to identify gene-gene relationships without framing such relationship into any biological concepts like gene regulatory network. Even though gene regulation is an important gene-gene relationship from transcription profile, there could be other subtle signals of gene-gene interaction beyond gene regulation. Therefore, the scope and conceptual framework of our work is different from those works.
> 2. SCENIC, GRNBoost2, SCODE and SCRIBE takes traditional statistical or machine learning approaches to discover gene regulatory networks. On the other hand, our method is based on modern deep learning framework, the Transformer architecture, with specific emphasis on attention between features. We believe this attention focus aligns very well for the goal of discovering gene-gene interactions.
>
> For these reasons, we feel like **these applications are in a different arena** from ours in terms of identifying gene-gene relationships. Therefore, we didn’t include these works. In the future work, we can explicitly focus on gene regulatory network conceptual framework and make head-to-head comparison with these algorithms.

---

> > ### Comment · Reviewer_8Hw9 · 2024-12-01
> >
> > I had trouble figuring out which points in my review some of the above are in response to.  The distinction between finding gene-gene interactions and finding gene regulatory networks seems pretty minor to me.  Both induce a network on genes, so it seems sensible to compare the methods.

---

> ### Author Response · Authors · 2024-11-25
>
> # [W4 - Line 161 - Please see our clarification.]
> >In line 161, you say that you train a Transformer model to classify whether a cell is an "Alzheimer's disease-infected cell or not." First, Alzheimer's is not an infection. But more importantly, there is no way to label individual cells as being affected by Alzheimer's or not. I am guessing that this sentence should say that you are labeling cells based on whether they come from an individual with Alzheimer's disease. How exactly this is done should be clarified.
>
> Thank you very much for point this out.
> We have updated our statement regarding the disease, especially AD. We train a Transformer model to classify whether a cell is from an Alzheimer’s patient or a healthy individual. We revise the statement in **line 67-69, line 161, line 172, line 217, line 236, line 249** and **line 381** accordingly.
>
>
>
>
>
> # [W5 - Evaluation on gene-gene interaction - Please see our clarification.]
> >The evaluation of the gene-gene interactions is problematic. The approach amounts to filtering a large set of known gene-gene interactions to include only those interactions that are implicated in Alzheimer's disease. This ignores the fact that many genes not involved in Alzheimer's disease continue to function and interaction with one another in the cell. It's not actually clear to me whether the evaluation considers pairs of genes not involved in Alzheimer's. It seems, from the description, like these pairs are treated as non-interacting.
>
> We would like to make it clear that the gene-gene relationships are inferred from a Transformer model trained to differentiate cells obtained from Alzheimer’s patients versus healthy controls. Because the model learns to differentiate Alzheimer’s disease from healthy controls, only genes and gene-gene relationships that are significantly different between these two states are identified/stands out. Even though many other genes function and interact, but they behave the same in both Alzheimer’s cells and healthy cells, therefore are not picked up by the Transformer model since there is no contribution from these genes in terms of differentiating Alzheimer’s and normal cells.
> We have revised our manuscript to clarify this in **line 67-69, line 161, line 172, line 217, line 236, line 249** and **line 381**.

---

> ### Author Response · Authors · 2024-11-27
>
> # [Q1- Transformer for gene-gene interactions - We would like to highlight the imporatnce of gene-gene interaction and the limitation of existing pre-trained Transformers.]
> >Why develop a new transformer model if the goal is just to identify gene-gene interactions? It seems like it would be easier to work with some existing model.
>
>
> We wish to emphasize the distinction in research focus between AI and biomedical studies. Our goal is to leverage our developed method to advance the understanding of human diseases, addressing the fundamental challenge of uncovering disease mechanisms in biomedical research. Specifically, we trained our Transformer model on a disease-specific dataset to identify gene-gene relationships pertinent to Alzheimer’s disease.
>
> While we acknowledge the potential for applying our attention aggregation pipeline to other Transformer models, such as scGPT and scFoundation, as demonstrated in Table 2, we did not observe significant advantages from these foundation models for our specific task. Nonetheless, our computational framework remains generalizable, and we welcome the incorporation of additional pre-trained models for scRNA data in future explorations.
>
>
> # [Q2- Celluformer, scGPT and scFoundation - Our gene-gene interaction is generalizable to different transformer models. And we see the advantage of Celluformer directly trained on disease data.]
> > How does the proposed Transformer model differ from scGPT, GeneFormer, and scFoundation? On line 124 we are told that they are similar to CelluFormer, but I don't see any indication of how they differ.
>
> Our work focuses on developing a novel attention aggregation pipeline while analyzing Alzheimer’s disease datasets to gain insights into disease mechanisms. While fine-tuning pre-trained models like scGPT is feasible, these models are trained on data unrelated to Alzheimer’s disease, which may result in gene-gene relationships that lack relevance to this specific condition. We don't see advantages of pre-trained models in our task in **Table 2**.
>
> In contrast, by training our own Transformer-based model, Celluformer, exclusively on Alzheimer’s disease data, we ensure that the identified gene-gene relationships are directly tied to the disease. This targeted training avoids contamination from unrelated datasets, providing insights that are both precise and disease-specific.
>
> Pre-trained models are typically trained on vast and diverse datasets, producing aggregate gene-gene relationships that represent generalized summaries. However, because Celluformer is trained on Alzheimer’s disease versus healthy datasets, the relationships it identifies are uniquely reflective of the disease, free from influence by unrelated data sources.
>
>
> # [Q3- Clarification on scatter addition - Sure!]
> >How do the "scatter addition operations" described in line 231 work? This entire paragraph would benefit from a more formal treatment, since I found the textual description very hard to follow.
>
> As shown in **line 224-236** and **Figure 2**, running inference on a cell $x$ on the transformer will produce a sparse attention matrix $A$, where the non-zero entries represents the average of attention scores between two genes expressed in cell $x$ when it go through the transformer. We take the average of all sparse attention matrices obtained from each cell to represent gene-gene interactions we modeled from the model on the whole dataset $X$.
>
>
> # [Q4- Justification of Sampling - Sure!]
> >On the face of it, the fact that you can achieve the same performance from 1% of the data a...
>
>
> We appreciate the feedback provided. Our performance evaluation is centered on comparisons with existing methods, demonstrating that our approach significantly outperforms others when utilizing 100\% of the data (see Table 2). Building on this benchmark, we further explore ways to enhance data efficiency within this framework. In particular, we emphasize that **identifying a representative subset offers an effective strategy for leveraging large datasets**, which constitutes the primary contribution of this work.
>
>
> # [Q5- Details of cell types - Sure!]
> >How are the cell type labels in Table 1 derived? More detail needs to be given to make this experiment reproducible.
>
>
> The labels of the cell types are provided by data generator. Specifically, the labels are at two levels. First, cells are labeled by the sample origin, either from Alzheimer’s Disease patients or healthy control. Second, the transcription profiles of sequenced cells are analyzed by data generators and shared. Cells are clustered and assigned 24 different cell types based on clustering results and the cell type specific marker genes. Our analysis focused on 18 types of neuron cells as they are most relevant to the Alzheimer’s disease – a neural degenerative disease.  We have revised the manuscript to clarify these points on **line 398**.

---

> > ### Comment · Reviewer_8Hw9 · 2024-12-01
> >
> > For Q1, I don't think you have convincingly made progress in understanding AD.  This is an ML conference, so a paper focused on advancing our understanding of human disease would be a bit out of place.
> >
> > Q4 seems to re-emphasize that the primary contribution is identifying a representative subset.  As my review mentions, this is a well studied problem, and the method should be compared against existing methods for this task.

---

> > > ### Author Response · Authors · 2024-12-02
> > >
> > > We thank the reviewer for the comments. We have added the original comments to each bullet point in the reply for a clear reference.
> > > Please see the following clarification.
> > >
> > > ### [A paper focused on advancing our understanding of human disease would be a bit out of place - Our work falls into a listed topic in the CFP of ICLR, please clarify on the "out of place" statement]
> > > >For Q1, I don't think you have convincingly made progress in understanding AD. This is an ML conference, so a paper focused on advancing our understanding of human disease would be a bit out of place.
> > >
> > > To clarify, we respectfully refer to the Call for Papers for ICLR: *We consider a broad range of subject areas including feature learning, metric learning, compositional modeling, structured prediction, reinforcement learning, uncertainty quantification and issues regarding large-scale learning and non-convex optimization, as well as applications in vision, audio, speech, language, music, robotics, games, **healthcare, biology**, sustainability, economics, ethical considerations in ML, and others.*
> > > Moreover, *applications to physical sciences (physics, chemistry, **biology**, etc.)* is one of the topics listed in the call for papers.
> > > We respectfully ask for an open and thoughtful consideration of our paper, as it aligns with one of the listed topics for ICLR. Additionally, we kindly seek clarification regarding the statement that our paper is "out of place."
> > >
> > > ### [Comparison with existing sampling - We kindly request supporting evidence for existing sampling algorithms with $O(n)$ processing time, constant memory and constant-time sampling.]
> > > >Q4 seems to re-emphasize that the primary contribution is identifying a representative subset. As my review mentions, this is a well studied problem, and the method should be compared against existing methods for this task.
> > >
> > > We kindly request supporting evidence for existing diversified sampling algorithms that achieve $O(n)$ processing time and constant memory while enabling constant-time sampling. Given the rigorous nature of the ICLR review process, we respectfully seek examples of such baselines to improve our understanding and facilitate meaningful discussion.
> > >
> > > ### [Minor difference between methods based on gene regulatory network and scRNA data - Comparing these two methodologies is inherently problematic.]
> > > >Comment: I had trouble figuring out which points in my review some of the above are in response to. The distinction between finding gene-gene interactions and finding gene regulatory networks seems pretty minor to me. Both induce a network on genes, so it seems sensible to compare the methods.
> > >
> > >
> > > We would like to clarify a fundamental difference between the methods based on gene regulatory network and our method based on scRNA data: Network-based approaches rely on a conceptual framework that models a specific type of gene-gene interaction based on the concept of gene regulation (transcription factors and their target genes), so everything they predicted and generated are based on this conceptual framework. As a result, their gene-gene interactions are either a transcription factor regulating a gene (transcription factor – target gene pair) or a co-expression (two genes both are regulated by a common transcription factor). However, **there are many more gene-gene relationships that go beyond this framework**. That’s why we argue our method is: 1) data-driven without imposing any conceptual framework, and 2) our approach captures a more diverse range of gene-gene relationships. If we compare our results to these methods based on the gene regulatory network, there are several issues: 1) not all the gene-gene relationships we identified fit the gene regulation framework so they will be considered false positives in that framework; 2) the methods based on gene regulatory network is not designed to identify other gene-gene relationships that our methods can find. For these reasons, comparing these two methodologies is inherently problematic, akin to comparing apples to oranges.

---

> > > > ### Comment · Reviewer_8Hw9 · 2024-12-02
> > > >
> > > > Thanks for clarifying what parts of the review each of your points is responding to.
> > > >
> > > > I understand that your proposed sketching method is O(n), whereas existing ones are not.  But I still think that a comparison to those methods is in order.  The methods were referred to in my original review: (e.g., Hie et al., Cell Systems 2019; Yang et al., ACM-BCB 2020; Yi & Stanley, bioRxiv, 2023; Hao et al. Nat Biotech 2024).
> > > >
> > > > As for whether biological application papers are within scope, I totally agree that they are.  I'm just saying that the point of an application paper in ICLR would typically be to show a methodological, as opposed to biological, advance. Indeed, your abstract seems to frame the contribution as a methodological one, that offers a "scalable and efficient pathway to deeper biological insights," rather than offering such insights per se.  So to clarify my previous point, I was just saying that if your paper had been solely about using existing methods to explore biology, then it would have been out of place.  But it's not, so this line of discussion is a bit off point.
> > > >
> > > > Regarding gene regulatory networks, it is true that there are multiple types of gene-gene interactions.  For example, the gene products (proteins) can interact with each other, in which case it is protein-protein interaction, which also has a large literature.   It seems like by being inclusive about all types of interactions that you aim to discover, you end up not having to compare to any previous methods, which feels sort of odd to me.  Maybe it would help to clarify exactly what kinds of interactions go into BioGrid.

---

> > > > > ### Author Response · Authors · 2024-12-02
> > > > >
> > > > > We thank the reviewer for the prompt reply. Please see the following clarification.
> > > > >
> > > > > ### [Methods in the original review - Please find below a comprehensive explanation of each method]
> > > > > >“I understand that your proposed sketching method is O(n), whereas existing ones are not. But I still think that a comparison to those methods is in order. The methods were referred to in my original review: (e.g., Hie et al., Cell Systems 2019; Yang et al., ACM-BCB 2020; Yi & Stanley, bioRxiv, 2023; Hao et al. Nat Biotech 2024).”
> > > > >
> > > > > Hie et al, Cell Systems 2019 – This study focuses on scRNA-Seq datasets, specifically addressing cell types and states rather than gene-gene interactions. While it involves subsampling datasets, the problem context is distinct from our work. Moreover, the computational requirements of their method differ significantly. As noted in their paper (Page 15), the plaid covering algorithm has a time complexity of $O(n\log n)$ per dimension, primarily due to sorting, and space complexity of **$O(n)$**. This contrasts with our method, which maintains a constant memory footprint and avoids the multiplicative $O(\log n)$ term in time complexity.
> > > > >
> > > > > Yi & Stanley, bioRxiv, 2023 – This work addresses multi-cellular sample embeddings from single-cell data, which is unrelated to gene-gene interactions. Additionally, their methods include geometric sketching, kernel herding, and random sampling (Page 3). Geometric sketching has a similar complexity to Hie et al, Cell Systems 2019, while kernel herding involves **quadratic time** complexity with respect to $n$. These approaches are computationally more expensive than ours.
> > > > >
> > > > >
> > > > > Hao et al. Nat Biotech 2024 – This work utilize the goemetric sketching as Hie et al, Cell Systems 2019, which shares the same time and space extra overhead.
> > > > >
> > > > > Yang et al, ACM-BCB 2024 -  This paper does not include a formal time complexity analysis but relies on submodular optimization techniques. These methods have computational demands comparable to geometric sketching, making them less efficient than our approach.
> > > > >
> > > > > To the best of our knowledge, none of the methodologies studies our problem and achieves a comparable time and space complexity. As such, these methods are not suitable for direct comparison. We emphasize that our proposed method offers both computational efficiency and relevance to the task at hand.
> > > > >
> > > > >
> > > > > ### [Contribution in ML - We would like to find a synergy between ML algorithm and biological impact.]
> > > > > > As for whether biological application papers are within scope, I totally agree that they are. I'm just saying that the point of an application paper in ICLR would typically be to show a methodological, as opposed to biological, advance. Indeed, your abstract seems to frame the contribution as a methodological one, that offers a "scalable and efficient pathway to deeper biological insights," rather than offering such insights per se. So to clarify my previous point, I was just saying that if your paper had been solely about using existing methods to explore biology, then it would have been out of place. But it's not, so this line of discussion is a bit off point.
> > > > >
> > > > > We are pleased that the reviewer acknowledges the relevance of our work to ICLR's thematic areas. Our research represents a synergistic integration of algorithmic innovations, specifically our proposed method, WDS, with real-world applications that demonstrate significant biological impact.

---

> ### Author Response · Authors · 2024-11-30
>
> Dear Reviewer 8Hw9
>
> Thanks again for helping review our paper! Since we are approaching the end of the author-reviewer discussion period, would you please check our response and our revised paper and see if there is anything we could add to address your concerns?
>
> We appreciate the time and effort you spend reviewing our paper.

---

> ### Author Response · Authors · 2024-12-02
>
> ### [Clarification on the interactions go into BioGrid -  Please see the following clarificaiton]
> >Regarding gene regulatory networks, it is true that there are multiple types of gene-gene interactions. For example, the gene products (proteins) can interact with each other, in which case it is protein-protein interaction, which also has a large literature. It seems like by being inclusive about all types of interactions that you aim to discover, you end up not having to compare to any previous methods, which feels sort of odd to me. Maybe it would help to clarify exactly what kinds of interactions go into BioGrid.
>
> I am glad that this reviewer agrees with our points. Indeed, BioGRID is an open‐access database resource that houses manually curated protein and genetic interactions from multiple species including yeast, worm, fly, mouse, and human. However, even all types of interaction there may not cover all the possible gene-gene relationships. On the other hand, our approach is purely data-driven, that is, any gene pair that contributes significantly to differentiating AD from non-AD cells will be captured. Our contribution is to develop a true discovery tool for any gene-gene relationships by combining large amounts of data from scRNA-Seq data and the amazing ability of deep learning frameworks like Transformer that can pick up subtle signals from large amounts of data. Our evaluation did emphasize whether our model, through a purely data-driven approach, successfully uncovers gene-gene interactions supported by BioGRID.
>
> Regarding comparisons to other methods, we are not bypassing this step. Rather, the limited availability of directly comparable methods and literature makes a fair comparison challenging. Therefore, we benchmark our approach against alternative methodologies addressing the same problem, as presented in Table 2.

---

### Official Review · Reviewer_QUgF · 2024-11-01

**Soundness:** 1
**Presentation:** 1
**Contribution:** 1
**Rating:** 3
**Confidence:** 2

**Summary:**

The paper propose a method that is related to transformer architecture to do gene-gene interaction discovery in single cell data. The key idea of the method is a density-based sampling method to reduce the data size.

Edit: after the rebuttal, I still keep the scores.

**Strengths:**

N/A

**Weaknesses:**

The paper has major problems that prevent me from understanding the method itself and its relations to related methods such as transformer and previous work on the same problem.
- Problem setting is not clear: It is stated that data set X given, so what is a disease D is in the dataset? What need to be learnt? How many dimensions are there in X, |V| or m?
- Model is not clear: what is "f"? The proposed model is not described anywhere in the text. Where is the interaction map in the model? What does it mean by "f" can successfully predict a disease infection? What does it mean by a gene pair that contribute the most to "f"?
- I think the paper simply refers the model to transformer architecture but the data here vectors!?!
- Conditions 2.1 on permutation invariant is totally misleading. "f" is defined for vectors, should we permute vector elements?
- While the data is said to be sparse, there are up to ... 12000 expressed genes in a cell, most of them have 2000+ expressed genes?
- The definition of Min-Max density is clumsy since it can refer to its root: kernel density estimation.

**Questions:**

Refer to the weaknesses for the questions that need to be addressed.
More general ones:
- What are the related methods for this problem?
- What is the model proposed here?

---

> ### Author Response · Authors · 2024-11-25
>
> We thank the reviewer for the comments. Please see the following clarifications.
>
>
> # [W1 - Clarification on data settings.]
>
> In our study, $D$ represents the disease of interest, such as Alzheimer's disease (AD). The goal is to develop a model that classifies whether a cell originates from an AD-origin individual or not. Subsequently, we use the model's attention mechanisms to uncover gene-gene interactions.
>
> The SEA-AD dataset comprises 1,240,908 cells, meaning the dataset $X$ includes 1,240,908 elements. Each cell's gene expression data is drawn from a total of 36,601 genes, denoted as $V$. For each cell, the number of expressed genes, represented as $m$ varies from 2000 to 5000. Consequently, each element in $X$ is a sparse feature vector with $V = 36,601$ dimensions and up to $m$ non-zero values.
>
> We have included a detail introduction of our data in **line 116-120**.
> For further details about our dataset and experimental setup, we refer reviewers to Sections 2.1 and 4.1 of the manuscript.
>
> # [W2 - Clarification on model settings.]
>
> We use $f$ to represent the Transformer model used to classify whether a cell $x \in X$ originates from an individual with Alzheimer's Disease (AD) or from a healthy person. The self-attention mechanism in $f$ captures pairwise interactions between expressed genes in cell $x$.
> We hypothesize that if the model $f$ accurately performs the proposed classification task, gene pairs with high interaction values across different attention blocks in $f$ could serve as indicators of novel gene-gene interactions associated with AD, as such gene pair contributes significantly in differentiating AD cells from non-AD cells.
>
> We have modified the manuscript to address this concerns on **line 161, line 172**.
>
>
> # [W3 & W5 - Clarification on Tokenziation of Gene Expressions.]
>
>
> We clarify the tokenization technique used for representing gene expressions in a cell $x$, considering two perspectives:
>
> 1. $x$ is represented as a high-dimensional, sparse, non-binary vector, where each dimension corresponds to a gene. Non-zero entries indicate the genes expressed in this cell, and their values represent the respective expression levels.
> 2. Alternatively, $x$ can be represented as a set of tuples $(\text{id}, \text{value})$, where $\text{id}$ denotes the gene expressed in cell $x$, and $\text{value}$ represents the expression level.
>
>
> Yes, the model in our work is a transformer architecture.
>
> In our approach, we adopt the second perspective. A cell is represented as a set of tuples $(i, v_i)$ where $i$ is the ith gene and $v_i$ is the expression value of that gene. In the model , we utilize an embedding table to map each gene to a vector representation. For each cell, the retrieved vectors from the embedding table are scaled by their corresponding expression levels, resulting in contextualized gene representations specific to the cell.
>
>
>
> Human genome has over 25,000 genes. However, because each cell has very little mRNA expressed for each gene, the detection rate of these expressed genes is low, ranging from 10-20% (2000-5000 genes) in most cells, but there could be few cells with higher number of genes measured than the range above. We have modified the manuscript on **line 149** to clarify this point.
>
>
>
> # [W4 - Clarification on permutation invariance.]
>
>
> We represent cells as a set of tuples (gene id and their expression value) rather than as vectors, introducing the concept of permutation invariance based on our interpretation of a cell as an unordered collection of tuples. This approach acknowledges that the sequential order of these tuples may vary due to differences in sequencing techniques. By treating the tuples as a set rather than a sequence, we ensure that the model avoids introducing positional bias into the representation.
>
>
> This tuple-based representation, along with its implications for permutation invariance, is further clarified in **line 116**.
>
>
> # [W6 - Clarification on Min-Max Kernel Density.]
>
> We have simplified our Min-Max density definitions in **line 291**.
> The Min-Max kernel stands out from general kernels due to its ability to be efficiently linearized using probabilistic hashing algorithms. This feature serves as the algorithmic foundation of our approach. By applying a low-cost computational procedure, we hash each element of the massive dataset once, enabling the use of count statistics for kernel density estimation.

---

> > ### Comment · Reviewer_QUgF · 2024-12-03
> >
> > Thank you for the clarification.
> >
> > I still find the paper, after clarification, making trivial concepts complicated. The tokenization process is no more than a sparse vector representation. The permutation invariance properties (Condition 2.1) is a little technical property (rather trivial) of functions acting on this representations, NOT the property of the functions acting on vectors. The part representing sets of cell is not related to this part. The comparison with Transformer not satisfying the condition is meaningless.
> >
> > I still find the paper as I evaluated after clarification, hence, keeping my scores unchanged.

---

> ### Author Response · Authors · 2024-11-30
>
> Dear Reviewer QUgF
>
> Thanks again for helping review our paper! Since we are approaching the end of the author-reviewer discussion period, would you please check our response and our revised paper and see if there is anything we could add to address your concerns?
>
> We appreciate the time and effort you spend reviewing our paper and the suggestions you have provided!

---

### Official Review · Reviewer_MPdJ · 2024-11-02

**Soundness:** 3
**Presentation:** 3
**Contribution:** 2
**Rating:** 5
**Confidence:** 4

**Summary:**

This paper introduces a novel computational framework designed to discover gene-gene interactions linked to complex diseases through single-cell transcriptomic data. Utilizing a Transformer model named CelluFormer, the authors address the challenge of computational efficiency by implementing a weighted diversified sampling algorithm. This algorithm allows the selection of a representative data subset by calculating a diversity score based on the Min-Max density kernel.

**Strengths:**

- The proposed method significantly reduces computational requirements without sacrificing accuracy. This approach addresses a key challenge in handling large-scale single-cell transcriptomic data.
- By leveraging Transformer models (CelluFormer) for gene-gene interaction discovery, the paper effectively adapts state-of-the-art NLP techniques to bioinformatics.
- The extensive experimental validation across multiple datasets and comparison with various baselines (e.g., Pearson Correlation, Spearman’s Correlation) provides empirical support for the proposed method’s effectiveness and robustness in data efficiency.

**Weaknesses:**

- The paper does not thoroughly discuss potential limitations or biases in using Transformer attention maps for gene-gene interaction discovery, such as how model-specific patterns may impact biological interpretability or generalizability.
- While the diversity score and sampling algorithm are well-motivated, the paper lacks detailed explanations regarding parameter sensitivity (e.g., choice of sample size) and the scalability of the method to even larger datasets or different cell types beyond the focus on Alzheimer’s Disease data.
- The paper could benefit from a more comprehensive comparison with existing gene interaction discovery techniques, especially non-Transformer-based methods that might offer complementary insights or efficiency advantages.
- For the scGPT model, there are cases where it performs better than the proposed method on specific datasets. Therefore, simply attributing the foundation model's lower performance to overfitting to pretrained knowledge or a mismatch between pretraining and fine-tuning data seems insufficient to support the claim that it underperforms compared to the proposed method.

**Questions:**

Please see the weaknesses.

---

> ### Author Response · Authors · 2024-11-25
>
> We thank the reviewers for the comments and suggestions to improve our paper. Please see the following clarification.
>
> # [W1 - Model-specific patterns of Transformers - Our experiments do observe the impact of variations in transformer architectures!]
> > The paper does not thoroughly discuss potential limitations or biases in using Transformer attention maps for gene-gene interaction discovery, such as how model-specific patterns may impact biological interpretability or generalizability.
>
> Transformer-based models with self-attention mechanisms excel at capturing pairwise interactions between genes via their attention maps. However, we do see that variations in Transformer architectures influence how gene-gene interactions are modeled. For example, unlike Celluformer that does not introduce any specific structure, scFoundation [1] employs the kernel-based approximation Transformer variant, Performer [2], in its decoder. Notably, using scFoundation's attention maps for gene-gene interactions results in a performance decline in GSEA analysis. We will include a detailed analysis in the updated version of the paper.
>
>
>
> # [W2 - Adaptivity of sampling methods - The exepected effective sample size is at $O(\log^2(n))$ scale.]
> >While the diversity score and sampling algorithm are well-motivated, the paper lacks detailed explanations regarding parameter sensitivity (e.g., choice of sample size) and the scalability of the method to even larger datasets or different cell types beyond the focus on Alzheimer’s Disease data.
>
> The weighted diversified sampling approach introduced in this work builds on theoretical foundations related to local density estimation [3]. According to this analysis, the sample size required to approximate the target kernel density with a multiplicative error of $\epsilon$ and failure probabiltiy $\delta$ bounded by $O(\log^2(n)\cdot \log(1/\delta)/\epsilon^2)$, where $n$ represents the total number of elements in the dataset. We have included a discussion in **lines 345–350** analyzing the sample size required to estimate the interaction score using the proposed weighted diversified sampling approach.
>
>
> # [W3 - Comparison with non-Transformer-based approaches - We have included multiple baselines with our available computational resources.]
> >The paper could benefit from a more comprehensive comparison with existing gene interaction discovery techniques, especially non-Transformer-based methods that might offer complementary insights or efficiency advantages.
>
> Table 2 presents the evaluation of our approach alongside non-transformer-based methods, including NID, Pearson, CS-CORE, and Spearman. Additional baselines, such as locCSN and SpQN, were not included due to their high computational cost, which exceeds the resources outlined in Appendix C.1.
>
>
> # [W4 - Analysis of scGPT performance - We agree that there may be multiple reasons.]
> > For the scGPT model, there are cases where it performs better than the proposed method on specific datasets. Therefore, simply attributing the foundation model's lower performance to overfitting to pretrained knowledge or a mismatch between pretraining and fine-tuning data seems insufficient to support the claim that it underperforms compared to the proposed method.
>
> We appreciate the reviewer's insightful assessment. The data handling approaches of foundation models, such as using rank instead of absolute expression values in scGPT, combined with the vast datasets used for training, make it challenging to isolate all factors contributing to the observed low performance. Potential influences may include differences in **gene vocabulary** and **local minima** arising from model training dynamics.
> In addition, variations in transformer model architecture could also contribute to the performance bias and variation as suggested by W1.
>
>
> We have revised the claims and discussion in **lines 430–444** to address potential reasons why foundation models have not demonstrated significant advancements in capturing gene-gene interactions. Future research could focus on a comprehensive evaluation of these factors to better understand their impact on model performance, particularly in identifying gene-gene interactions. Emphasis should be placed on the role of gene vocabulary, the effects of training dynamics and the impact of model variation.
>
>
> [1] Large-scale foundation model on single-cell transcriptomics
> [2] Rethinking Attention with Performers
> [3] Local Density Estimation in High Dimensions

---

> > ### Comment · Reviewer_MPdJ · 2024-11-26
> >
> > Thank you for your detailed responses and for addressing the comments provided. I appreciate the effort you have put into revising the manuscript and clarifying the points raised. However, there are a few areas that I believe could benefit from additional discussion or exploration, which I outline below.
> >
> > - [Related to W1] Thank you for addressing the impact of Transformer variations and planning to include a detailed analysis. Could you kindly elaborate on how the model-specific patterns in Celluformer might influence biological interpretability or generalizability?
> >
> > - [Related to W3] I apologize for the vague wording. From my understanding, the methods compared in Table 2 can be broadly categorized as: 1) statistical approaches, 2) MLP-based NID, and 3) the scFoundation model (please correct me if I am mistaken). Within these categories, I feel that there is a lack of deep learning methods (e.g., GNNs) or machine learning techniques (e.g., SVM, RF) for feature selection and gene interaction discovery.
> >
> > - [Related to W4] The authors stated in the manuscript:
> > >"In addition, the foundation models, scGPT and scFoundation, achieved comparable performances with other baselines across all datasets."
> > >"We attribute this outcome to two main factors: (1) Overfitting to Pretrained Knowledge and (2) Mismatch Between Pretraining and Fine-Tuning Data."
> >
> >   However, in the rebuttal, it was mentioned that
> >   >"Potential influences may include differences in gene vocabulary and local minima arising from model training dynamics."
> >
> >   Does this imply that there is no concrete support for the two main factors claimed in the manuscript?

---

> > > ### Author Response · Authors · 2024-11-29
> > > **Thank you for your engagement! Please the the following clarification.**
> > >
> > > ### [Response to W1]
> > > >[Related to W1] Thank you for addressing the impact of Transformer variations and planning to include a detailed analysis. Could you kindly elaborate on how the model-specific patterns in Celluformer might influence biological interpretability or generalizability?
> > >
> > > At the **model level**, CelluFormer employs all-against-all self-attention to capture gene-gene interactions comprehensively. This design ensures that the architecture of CelluFormer itself does not impose model-specific patterns, as its primary function is to extract interactions without introducing biases tied to the model structure.
> > >
> > > At the **data level**, we utilize CelluFormer as a specialized analysis tool to uncover gene-gene interactions specific to a given dataset. This approach means that CelluFormer is not intended for cross-dataset generalization; instead, a separate CelluFormer model is trained for each dataset to ensure the interactions identified are directly relevant to the biological context of that dataset. Consequently, any model-specific patterns are inherently tied to the data they are trained on, ensuring that interpretability remains dataset-specific. Importantly, the interpretability derived from one dataset does not influence or compromise the interpretability of another.
> > >
> > >
> > >
> > > In our study, we applied CelluFormer across seven distinct neuron cell types. The model demonstrated consistent performance across these datasets, as reflected in the F1 scores and accuracy metrics (**Table 1**). This consistency supports the generalizability of CelluFormer across diverse cell types. Additionally, our results indicate that the robustness and generalizability of the model improve significantly with larger training datasets.
> > >
> > >
> > >
> > > Furthermore, training CelluFormer on datasets representing multiple cell types with a consistent labeling scheme (in our case, seven neuron cell types labeled as either AD or non-AD) led to substantial improvements in performance. This approach not only enhanced the model's ability to identify gene-gene interactions for specific labels but also revealed interactions associated with Alzheimer’s disease conditions (**Table 1**).
> > >
> > >
> > >
> > > In summary, CelluFormer offers a powerful, data-specific approach to uncovering gene-gene interactions, with demonstrated robustness, generalizability, and potential for improved insights when leveraging diverse and well-labeled training datasets.

---

> > > > ### Author Response · Authors · 2024-11-29
> > > >
> > > > ### [Response to W3]
> > > >
> > > > >[Related to W3] I apologize for the vague wording. From my understanding, the methods compared in Table 2 can be broadly categorized as: 1) statistical approaches, 2) MLP-based NID, and 3) the scFoundation model (please correct me if I am mistaken). Within these categories, I feel that there is a lack of deep learning methods (e.g., GNNs) or machine learning techniques (e.g., SVM, RF) for feature selection and gene interaction discovery.
> > > > >
> > > > We thank the reviewer for their valuable suggestions and would like to address this question from the following perspectives.
> > > >
> > > > Our approach identifies gene-gene relationships using a data-driven methodology, without relying on predefined biological frameworks such as gene regulatory networks. In contrast, GNN-based methods [1, 2, 3, 4] depend on pre-constructed gene-gene interaction networks, which are built using prior biological knowledge. Additionally, we note that some SVM-based approaches [5, 6, 7] are designed for single nucleotide polymorphism (SNP) data, differing significantly from the single-cell RNA data analyzed in our study.
> > > >
> > > > Therefore, both GNN- and SVM-based methods are not directly applicable to single-cell data. We have revised the manuscript (**lines 868–884**) to include a detailed discussion of existing approaches for gene-gene interaction analysis.
> > > >
> > > >
> > > > [1] [Associating Genes and Protein Complexes with Disease via Network Propagation](https://journals.plos.org/ploscompbiol/article?id=10.1371/journal.pcbi.1000641)
> > > > [2] [Vavien: An Algorithm for Prioritizing Candidate Disease Genes Based on Topological Similarity of Proteins in Interaction Networks](https://pmc.ncbi.nlm.nih.gov/articles/PMC3216100/)
> > > > [3] [DeepDRIM: a deep neural network to reconstruct cell-type-specific gene regulatory network using single-cell RNA-seq data](https://academic.oup.com/bib/article/22/6/bbab325/6356429?login=true)
> > > > [4] [Deep learning for inferring gene relationships from single-cell expression data](https://www.pnas.org/doi/abs/10.1073/pnas.1911536116)
> > > > [5] [Detecting gene-gene interactions using support vector machines with L1 penalty](https://ieeexplore.ieee.org/document/5703819)
> > > > [6] [Gene-Gene Interaction Tests Using SVM and Neural Network Modeling](https://ieeexplore.ieee.org/document/4221209)
> > > > [7] [A support vector machine approach for detecting gene-gene interaction](https://pubmed.ncbi.nlm.nih.gov/17968988/)
> > > >
> > > >
> > > > ### [Response to W4]
> > > > >The authors stated in the manuscript:
> > > > "In addition, the foundation models, scGPT and scFoundation, achieved comparable performances with other baselines across all datasets." "We attribute this outcome to two main factors: (1) Overfitting to Pretrained Knowledge and (2) Mismatch Between Pretraining and Fine-Tuning Data."
> > > > >Does this imply that there is no concrete support for the two main factors claimed in the manuscript?
> > > >
> > > > To clatifiy more on the performance of scGPT an scFoundation, we have revised **line 430-444** for a detailed dicussion.  Especially, we state that "*The performance of foundation models like scGPT and scFoundation may stem from various factors. For instance, the data handling approaches of foundation models, such as using rank instead of absolute expression values in scGPT, combined with the vast datasets used for training, make it challenging to isolate all factors contributing to the observed lower performance.*"

---

> > > > > ### Comment · Reviewer_MPdJ · 2024-12-02
> > > > >
> > > > > **[Related to W1]**
> > > > >
> > > > > Thank you for your detailed response and for addressing the role of CelluFormer in capturing gene-gene interactions. While I appreciate the insights provided, I would like to understand more clearly how the approach in CelluFormer differentiates itself from the self-attention mechanism in standard Transformers. Specifically, could you elaborate on the unique aspects of CelluFormer’s gene-gene interaction modeling that contribute to its ability to uncover biologically relevant interactions? This clarification would help in distinguishing the model’s strengths in terms of biological interpretability and generalizability from existing Transformer-based methodologies.
> > > > >
> > > > > **[Related to W3]**
> > > > > Thank you for the detailed response and clarification. While I understand the rationale behind your argument that GNN- and SVM-based methods may not be directly applicable to single-cell RNA data due to differences in prior biological assumptions or data structure, I would like to respectfully share a differing perspective. Recent developments [1~6] have shown that both GNNs and SVMs can be adapted to analyze single-cell RNA data through modifications in preprocessing or network construction, even without relying on predefined gene-gene interaction networks [7,8]. For instance, GNNs can utilize correlations or co-expression patterns derived directly from single-cell data to infer relationships.
> > > > >
> > > > > I believe discussing such adaptations could provide a more balanced comparison and highlight the unique contributions of your approach in contrast to these methodologies. Including this perspective may also help readers who are familiar with these methods better appreciate the strengths of your work.
> > > > >
> > > > > -----
> > > > > [1] Zhang, W., Meng, Z., Wang, D., Wu, M., Liu, K., Zhou, Y., & Xiao, M. (2024). Enhanced Gene Selection in Single-Cell Genomics: Pre-Filtering Synergy and Reinforced Optimization. arXiv preprint arXiv:2406.07418.
> > > > >
> > > > > [2] Elyanow, R., Dumitrascu, B., Engelhardt, B. E., & Raphael, B. J. (2020). netNMF-sc: leveraging gene–gene interactions for imputation and dimensionality reduction in single-cell expression analysis. Genome research, 30(2), 195-204.
> > > > >
> > > > > [3] Wei, Q., Islam, M. T., Zhou, Y., & Xing, L. (2024). Self-supervised deep learning of gene–gene interactions for improved gene expression recovery. Briefings in Bioinformatics, 25(2), bbae031.
> > > > >
> > > > > [4] Wang, T., Bai, J., & Nabavi, S. (2021). Single-cell classification using graph convolutional networks. BMC bioinformatics, 22, 1-23.
> > > > >
> > > > > [5] Kim, J., Rothová, M. M., Madan, E., Rhee, S., Weng, G., Palma, A. M., ... & Won, K. J. (2023). Neighbor-specific gene expression revealed from physically interacting cells during mouse embryonic development. Proceedings of the National Academy of Sciences, 120(2), e2205371120.
> > > > >
> > > > > [6] Kommu, S., Wang, Y., Wang, Y., & Wang, X. (2024). Gene Regulatory Network Inference from Pre-trained Single-Cell Transcriptomics Transformer with Joint Graph Learning. arXiv preprint arXiv:2407.18181.
> > > > >
> > > > > [7] Wang, J., Ma, A., Chang, Y., Gong, J., Jiang, Y., Qi, R., ... & Xu, D. (2021). scGNN is a novel graph neural network framework for single-cell RNA-Seq analyses. Nature communications, 12(1), 1882.
> > > > >
> > > > > [8] Liu, T., Wang, Y., Ying, R., & Zhao, H. (2024). MuSe-GNN: learning unified gene representation from multimodal biological graph data. Advances in neural information processing systems, 36.

---

> > > > > > ### Author Response · Authors · 2024-12-02
> > > > > >
> > > > > > We thank the reviewer for the follow-up suggestions. Please see the following clarification.
> > > > > >
> > > > > > ### Further Clarification on CelluFormer
> > > > > > >[Related to W1]
> > > > > > >Thank you for your detailed response and for addressing the role of CelluFormer in capturing gene-gene interactions. While I appreciate the insights provided, I would like to understand more clearly how the approach in CelluFormer differentiates itself from the self-attention mechanism in standard Transformers.
> > > > > >
> > > > > > CelluFormer does not incorporate position bias, such as the positional encoding used in traditional transformers, because single-cell data comprises sets of gene expressions rather than sequential information. Additionally, as illustrated in Figure 2, for each input cell, CelluFormer retrieves the embeddings for individual genes. It then multiplies the expression level of each gene to the retrieved embedding. This approach differs from scFoundation, which employs both value and gene embeddings and combines them through element-wise summation. We will highlight this in Section 2.2.
> > > > > >
> > > > > >
> > > > > > >Specifically, could you elaborate on the unique aspects of CelluFormer’s gene-gene interaction modeling that contribute to its ability to uncover biologically relevant interactions? This clarification would help in distinguishing the model’s strengths in terms of biological interpretability and generalizability from existing Transformer-based methodologies.
> > > > > >
> > > > > > Proper normalization of the average attention scores between gene pairs in CelluFormer is critical when using large-scale single-cell datasets. For the full dataset, the average attention score for a gene pair should be calculated across all occurrences of that pair in the model. When employing WDS, a weighted average should be used, where the sample probability, $\mathcal{I}(x)$, serves as the weight (see **lines 351-355**).
> > > > > >
> > > > > >
> > > > > >
> > > > > >
> > > > > >
> > > > > >
> > > > > > ### Clarification on the literature study
> > > > > > >[Related to W3]
> > > > > > Thank you for the detailed response and clarification. While I understand the rationale behind your argument that GNN- and SVM-based methods may not be directly applicable to single-cell RNA data due to differences in prior biological assumptions or data structure, I would like to respectfully share a differing perspective. Recent developments [1~6] have shown that both GNNs and SVMs can be adapted to analyze single-cell RNA data through modifications in preprocessing or network construction, even without relying on predefined gene-gene interaction networks [7,8]. For instance, GNNs can utilize correlations or co-expression patterns derived directly from single-cell data to infer relationships.
> > > > > > I believe discussing such adaptations could provide a more balanced comparison and highlight the unique contributions of your approach in contrast to these methodologies. Including this perspective may also help readers who are familiar with these methods better appreciate the strengths of your work.
> > > > > >
> > > > > >
> > > > > > We agree with the reviewer's comments that multiple efforts, including the papers referred by the comments,  are developed to analyze scRNA-Seq data, including GNN and SVM-based approaches to identify gene-gene interactions. On the other hand, we would like to highlight that our work introduces a novel perspective by exploring the internal states of deep learning models to infer previously unrecognized gene-gene interactions. This methodology represents a shift from traditional data analysis techniques to a "discovery from models trained on data" paradigm, which we believe marks a significant step forward in advancing how we analyze scRNA-Seq data.

---

> > > > > > > ### Comment · Reviewer_MPdJ · 2024-12-03
> > > > > > >
> > > > > > > Thank you to the authors for their thorough responses and detailed clarifications. I appreciate the potential of CelluFormer in addressing limitations of existing Transformer-based models for single-cell RNA-seq data analysis. However, I believe the manuscript could benefit from addressing the following points in more depth:
> > > > > > >
> > > > > > > First, regarding the input encoding for scRNA-seq data, while the authors claim that CelluFormer has advantages over traditional Transformer-based models, this should be substantiated with more comprehensive methodological descriptions and experimental results. It would also be helpful to include a clear discussion on how these advantages translate into improved performance compared to prior approaches.
> > > > > > >
> > > > > > > Second, the authors must convincingly demonstrate the model's ability to uncover unknown gene-gene interactions. If the strength of the proposed model is positioned as "discovery from models trained on data," it is crucial to accompany this claim with biologically meaningful interpretations that validate the model's findings. For example, the manuscript could benefit from explicitly highlighting how the discovered gene-gene interactions lead to novel biological insights.
> > > > > > >
> > > > > > > In summary, while I see the potential of this work, the concerns raised remain unaddressed. As such, I will keep my initial score unchanged.

---

### Author Response · Authors · 2024-11-27

## Revision Summary
We thank all the reviewers for the time and effort in helping us improve the quality of the paper.
We have updated the paper to incorporate constructive suggestions. We summarize the major changes:

- **Revision on abstract and introduction**  We have revised our abstract and introduction to add more biological context of our research (**line 33-35, line 40-43, line 45-50, line 67-69, line 90-93**)
- **Revision on data (To Reviewer QUgF)**  We have included a detail introduction of our data in **line 149**.
- **Revision on disease description (To Reviewer 8Hw9)**  We have updated our statement regarding the disease, especially AD. We train a Transformer model to classify whether a cell is from an Alzheimer’s Disease patient or a healthy individual. In **line 67-69, line 161, line 172, line 217, line 236, line 249** and **line 381**.
- **Additional discussion on sample size (To Reviewer MPdJ)** We have included a discussion in **lines 345–350** analyzing the sample size required to estimate the interaction score using the proposed weighted diversified sampling approach.
- **Revision on discussion of foundation models' performance (To Reviewer MPdJ)** We have revised the claims and discussion in **lines 430–444** to address potential reasons why foundation models have not demonstrated significant advancements in capturing gene-gene interactions.

---

### Meta-Review · Area_Chair_jPyj · 2024-12-17

**Metareview:**

This paper introduces a transformer model for identifying gene-gene interactions linked to complex diseases through single-cell transcriptomic data.

The reviewers found strengths in the idea of adapting state-of-the-art NLP models to the problem of gene-gene interactions.

Significant concerns were notes by some reviewers. The review raised concerns about the claim that CelluFormer has advantages over traditional transformer-based methods. The review also notes concerns about the targeting of the paper and biological and/or methodological significance. Concerns were also raised about the specifics of the type of gene-gene interactions. It's important to carefully scope the paper to ensure the claims rest firmly on the foundation of the evidence. Clarifying the types of interactions is important to ensure this happens. Reviews also pointed out key literature that should be noted in the paper.

The authors and reviewers engaged in an extensive dialogue which was beneficial to the review process and to future improvements to the manuscript. Given the concerns and after a thorough review of the paper and conversation in the review process, the paper seems to be below the bar of acceptance at this time.


It seems it would be good for the authors to follow MPdj's suggestion about the literature here.

Reviewer 8Hw had a productive dialogue with the authors and raised some important points about targeting of the paper and biological and/or methodological significance. I tend to agree with the reviewer and it seems the point is settled. Thanks for the discussion on that point.

The reviewer also made a point about specificity of gene-gene interactions. It's important to carefully scope the paper to ensure the claims rest firmly on the foundation of the evidence. Clarifying the types of interactions is important to ensure this happens.

 I tend to agree that further methodological description and empirical evidence would be needed to support that claim. I also tend to agree with the sentiment that while "GNN- and SVM-based methods may not be directly applicable to single-cell RNA data due to differences in prior biological assumptions or data structure", it may be going too far at this stage of the literature to dismiss them when there is abundant recent work in that direction.

**Additional Comments On Reviewer Discussion:**

The reviewers and authors engaged in a substantive discussion on the merits of the paper. The dialogue helped clear up confusion over specific comments and at the end of the discussion, the state of understanding of the paper seemed to be good.

It's important for the authors to communicate their work clearly and, unfortunately, it is not possible to control how each reader (reviewer or not) will interpret the narrative of the paper. Yet, the primary responsibility rests on the authors to strive for as much clarity as possible in their presentation. The dialogue between the reviewers and authors demonstrates the interest of both parties to make the best research possible available to the community in a way that will be understood and appreciated by all.

---

### Decision · Program_Chairs · 2025-01-22

Reject